



# Geometrical and optical properties of cirrus clouds in Barcelona, Spain: Analysis with the two-way transmittance method of 5 years of lidar measurements

Cristina Gil-Díaz[1], Michäel Sicard[1,2,3], Adolfo Comerón[1], Daniel Camilo Fortunato dos Santos Oliveira[1], Constantino Muñoz-Porcar[1], Alejandro Rodríguez-Gómez[1], Jasper R. Lewis[4], Ellsworth J. Welton[5], Simone Lolli[6,1]

[1]CommSensLab, Dept of Signal Theory and Communications, Universitat Politècncia de Catalunya (UPC), Barcelona, 08034, Spain

[2]Ciències i Tecnologies de l'Espai-Centre de Recerca de l'Aeronàutica i de l'Espai/Institut d'Estudis Espacials de Catalunya (CTE-CRAE/IEEC), Universitat Politècnica de Catalunya (UPC), Barcelona, 08034, Spain

[3]now at: Laboratoire de l'Atmosphère et des Cyclones, Université de La Réunion, Saint Denis, 97744, France

[4]Joint Center for Earth Systems Technology, University of Maryland, Baltimore County, Baltimore, Maryland

[5]NASA Goddard Space Flight Center, Greenbelt, Maryland

[6]CNR-IMAA, Italian National Research Council , 85050 Tito Scalo (PZ), Italy

**Correspondence:** Cristina Gil-Díaz (cristina.gil.diaz@upc.edu)

**Abstract.** In this paper a statistical study of cirrus geometrical and optical properties based on 5 years of continuous ground-based lidar measurements with the Barcelona (Spain) Micro Pulse Lidar (MPL) is analysed. First, a review of the literature on the two-way transmittance method is presented. This method is a well-known lidar inversion method used to retrieve the optical properties of an aerosol/cloud layer between two molecular (i.e. aerosol/cloud-free) regions below and above, without the need to make any a priori assumptions about their optical and/or microphysical properties. Second, a simple mathematical expression of the two-way transmittance method is proposed for both ground-based and spaceborne lidar systems. This approach of the method allows the retrieval of the cloud optical depth, the columnar cloud lidar ratio and the vertical profile of the cloud backscatter coefficient. The method is illustrated for a cirrus cloud using measurements from a ground-based MPL and from the spaceborne Cloud-Aerosol Lidar with Orthogonal Polarization (CALIOP). Third, the data base is then filtered with a cirrus identification criterion based on (and compared to) the literature using only lidar and radiosonde data. During the period from November 2018 to September 2022, 367 high-altitude cirrus clouds have been identified at 00 and 12 UTC, of which 203 were successfully inverted with the two-way transmittance method. The statistical results of these 203 high-altitude cirrus clouds show that the cloud thickness is 1.8 ± 1.1 km, the mid-cloud temperature is -51 ± 8 ℃ and linear cloud depolarization ratio is 0.32 ± 0.13. The application of the transmittance method yields an average cloud optical depth (COD) of 0.36 ± 0.45 and a mean lidar ratio of 30 ± 19 sr. It is observed that the highest occurrence of cirrus is in spring and the majority of cirrus clouds (48%) are visible (0.03 < COD < 0.3), followed by opaque (COD > 0.3) with a percentage of 38%. Together with results from other sites, a possible latitudinal dependence of lidar ratio is detected: the lidar ratio increases with increasing latitude. We also note that in Barcelona the COD correlates positively with the cloud base temperature, lidar ratio and linear cloud





depolarization ratio and negatively with the cloud base height. On the one hand, the decrease of the cloud base temperature and
COD associated to an increase of the cloud base height occurs because clouds located at higher altitudes are formed from air
masses with a lower water vapour content and, therefore, their geometric and optical thickness are smaller. On the other hand,
the lidar ratio increases with increasing cloud optical depth, as the complexity and diversity of ice crystal shapes increases, due
to collisions and turbulence. Lastly, the linear cloud depolarization ratio has a slightly positive tendency with the cloud optical
depth, because as the cloud optical depth increases, the number of ice crystals increases and, as a consequence, the randomly
aggregation of ice crystals within the cloud occurs more frequently, making ice crystals rougher and thus more depolarizing.

## 1 Introduction

Aerosol-radiation and aerosol-cloud interactions still drive large uncertainties in our estimates of climate change. The last IPCC
report states a global mean effective radiative forcing (ERF) of -0.6 to 0 Wm$^{-2}$ for aerosol-radiation interaction (ERFari) and
a range of -1.7 to -0.3 Wm$^{-2}$ for aerosol-cloud interaction (ERFaci) (IPCC, 2021). Compared to the previous report (IPCC,
2013), there has been an increase in the estimated magnitude of the total aerosol ERF associated with a reduction of its uncer-
tainty, supported by a combination of increased process-understanding and progress in modelling and observational analyses.
The magnitude of ERFari has also decreased while the magnitude of ERFaci has increased. In particular, the radiative effect
of high-altitude cirrus clouds, known to play a fundamental role in the global radiation budget (Liou, 1986; Lolli et al., 2017),
have been designated as poorly understood by (IPCC, 2021) because of a lack of knowledge of their dynamic, microphysical
and radiative properties. Indeed, cirrus cloud critical role in the climate comes from the fact that 1) they are the only cloud that
can readily cool or warm the top of atmosphere and the surface, during daytime, depending on their properties (Campbell et al.,
2016) and 2) they have a high occurrence frequency globally (Holz et al., 2008). In fact, (Campbell et al., 2016) demonstrated
through a one-year long lidar dataset that positive or negative daytime cirrus cloud forcing could occur depending on the cloud
optical depth and the solar zenith angle. All these results call for more investigation on the cirrus cloud properties and their 3D
spatial distribution at the global scale.

The Met Office (the national meteorological service for the United Kingdom; https://www.metoffice.gov.uk/) defines cirrus
clouds as "short, detached, hair-like clouds found at high altitudes". Cirrus clouds are made of ice crystals and are often seen
during fair weather. Cirrus clouds can form by different atmospheric mechanisms, giving rise to cirrus clouds with different
physical, geometrical and optical properties. In the mid-latitude regions, the most common atmospheric mechanisms for cirrus
cloud formation are the deep convective outflow (Li et al., 2005; Fu et al., 2006; Jin et al., 2006), the large-scale uplift of hu-
mid layers induced by the Asian monsoon (Chen and Liu, 2005), and the cooling associated to the wave activity in the upper
troposphere (Spichtinger et al., 2003). Therefore, the atmospheric mechanisms of cirrus formation govern the type of cirrus
formed. For example, sub-visible cirrus clouds (COD < 0.03) are formed because of the cooling near tropopause height while
opaque cirrus are generally formed by deep convective outflow at lower heights except during deep overshooting convections





(Pandit et al., 2015).

Cirrus clouds can be characterized geometrically, physically and optically by some key parameters such as the mid-cloud altitude, mid-cloud temperature, cloud extinction coefficient, cloud optical depth, lidar ratio (LR), linear cloud depolarization ratio (LCDR) or cloud phase. While the LR and LCDR are related with the microphysical properties of the ice crystals contained in cirrus clouds, such as their shape and/or orientation, the mid-cloud altitude and temperature as well as the cloud extinction coefficient play an important role in determining the cloud radiative properties. Up to the present date, there is no exact theoretical solution for scattering and absorption by non-spherical ice particles (Liou and Takano, 1994). Nevertheless, scattering models for cirrus clouds have been developed, such as (Baran et al, 2009, 2011a, b) which relates the cirrus ice water content and mid-cloud temperature with its extinction coefficient, single scattering albedo and asymmetry factor. The main advantage of this model is that it is not necessary to calculate the ice particle shapes and their size distributions in order to calculate their radiative properties. Alternatively, (Heymsfield et al., 2014; Dolinar et al., 2022) propose to calculate the cirrus ice water content from the extinction coefficient at a visible wavelength and the effective geometric diameter of the ice crystals, which in turn is a function of temperature. Once the cirrus ice content and the effective geometric diameter of the ice crystals are obtained, the scattering and absorption coefficients and the asymmetry factor can be calculated with the (Fu et al., 1998, 1999) parametrizations.

Lidar systems are the only remote sensing instrument able to measure simultaneously vertical profiles of extinction and temperature. However most of single wavelength elastic lidars are not equipped with the technique for temperature detection (in general the integration lidar tehcnique or the rotational Raman technique; see (Behrendt, 2005)). In such cases, radiosoundings, when available, can provide the temperature measurements (Sassen, 1991). Although cirrus clouds are not their primary target, many projects/networks/instruments worldwide are capable of measuring cirrus extinction (or a good guess of it) from the ground: the European Aerosol Research LIdar NETwork, EARLINET (Pappalardo et al., 2014) now included in the Aerosols, Clouds and Trace gases Research Infrastructure, ACTRIS (Saponaro et al., 2019), Micro Pulse Lidar NETwork, MPLNET (Welton et al., 2001); and from space: Cloud-Aerosol Lidar and Infrared Pathfinder Satellite Observations, CALIPSO (Winker et al., 2007), AEOLUS (Ingmann and Straume, 2016), Earth Cloud, Aerosol and Radiation Explorer, EarthCARE (Eisinger et al, 2017).

The objective of this paper is to show a statistical analysis os cirrus cloud properties based on 5 years of continuous ground-based lidar measurements obtained from NASA Micropulse lidar network (MPLNET, https://mplnet.gsfc.nasa.gov/) ans meteorological profiles from radiosondes in Barcelona. Specifically, the daytime and nighttime cirrus geometrical (cirrus base and top height and thickness), thermal (temperature at base/mid/top cloud altitude) and optical properties (cloud optical depth, lidar ratio and linear cloud depolarization ratio) are investigated. The instrumentation used is presented in Section 2. A review and a new and unified formulation of the two-way transmittance method for both ground-based and spaceborne lidar systems is given in Section 3. Geometrical and optical cirrus properties are analysed in Section 4 and conclusions are presented in Section 5.





## 2   Instrumentation

Five years (2018 to 2022) of continuous lidar measurements performed with the MPL in Barcelona, northeast of Spain, are used in this paper. Co-located radiosoundings launched by the Meteorological Service of Catalonia (Meteocat) at 00 and 12 UTC are used as well. For the application of the two-way transmittance method for a high-altitude cirrus scene measured from

a spaceborne lidar system, data from CALIPSO satellite has been also used.

### 2.1   The MPL lidar

The NASA Micro-Pulse Lidar Network is a federated network of Micro-Pulse Lidar systems designed to measure aerosol and cloud vertical structure, and boundary layer heights (Welton et al., 2001). All sites in MPLNET currently use the MPL, which was developed at NASA Goddard Space Flight Center (GSFC) in the early 1990s. The MPL was patented and subsequently

licensed to industry for commercial sales beginning in the mid 1990s. The data collected by MPL instruments are continuously, day and night, over long time periods from sites around the world. Most MPLNET sites are co-located with sites in the NASA Aerosol Robotic Network (AERONET). MPLNET data have contributed to many studies and applications, such as: domestic and international aerosol and cloud research (Welton et al., 2000, 2002), climate change and air quality studies (Miller et al., 2011), support for NASA satellite and sub-orbital missions and aerosol modeling and forecasting (Misra et al., 2012).


The lidar system used in this study is a Polarized Micro Pulse Lidar (P-MPL) system that is integrated in the NASA Micropulse lidar Network. The Barcelona MPL is located on the roof of the CommSensLab (https://ors.upc.edu/) building in the Campus Nord of the Universitat Politècnica de Catalunya (41.38°N, 2.11°E; 115 m a.s.l.), approximately at 1 km from Serra de Collserola and 7 km from the sea.


The MPL system consists of a compact, eye-safe lidar designed for full-time unattended operation (Spinhirne, 1993; Campbell et al., 2002; Flynn et al., 2007; Huang et al., 2010). It uses a pulsed solid-state laser emitting low laser pulse energy $\sim$ $6\mu$J at a wavelength of 532 nm and a pulse repetition frequency of 2500 Hz. As both transmitting and receiving optics, the system uses a co-axial "transceiver" design with a Cassegrain telescope. The MPL systems use an optical setup that consists in

an actively controlled liquid crystal retarder which makes the system capable to conduct polarization-sensitive measurements by alternating between two retardation states (Flynn et al., 2007), while that the polar and cross-polar signals are separately acquired and recorded. Additionally, the MPL systems have a narrow receiver field of view, approximately 100 $\mu$rad (Campbell et al., 2002). Therefore, in this study the multiple scattering effect is considered negligible (Platt, 1973; Platt et al., 2002; Lewis et al., 2016).


Data are centrally processed at NASA GSFC through MPLNET version 3 (V3, released in 2021) algorithm and level 1.5 (L15, near real time, quality assured) data (Welton et al., 2018). In particular, we used the MPLNET Normalized Relative Backscatter (NRB) product, provided with 1-min temporal resolution and at 75m vertical resolution. This product includes





correction of deadtime, darkcount, afterpulse, background, overlap (Campbell et al., 2002; Welton and Campbell, 2002) and

polarization calibration (Welton et al., 2018). Cloud base height and cloud top height, as well as cloud optical depth and extinction coefficient profiles, linear volume depolarization ratio and cloud phase belong to MPLNET Cloud (CLD) product, described by (Lewis et al., 2016, 2020). A multi-temporal averaging scheme is used to improve high-altitude cloud detection under conditions of a weak signal-to-noise ratio by combining NRB signal profiles averaged to short (1-min), medium (5-min), and long (21-min) temporal resolutions into a merged cloud scene.

## 2.2 The CALIOP lidar

The Cloud-Aerosol Lidar and Infrared Pathfinder Satellite Observation (CALIPSO) satellite provides new insight into the role that clouds and atmospheric aerosols play in regulating Earth's weather, climate, and air quality, through the analysis of their vertical structure and properties (Sourdeval et al., 2012; Rita et al., 2016; Granados-Muñoz et al., 2019). CALIPSO is composed by three co-aligned nadir-viewing instruments: the Cloud-Aerosol Lidar with Orthogonal Polarization (CALIOP), the

Imaging Infrared Radiometer (IIR) and the Wide Field Camera (WFC). CALIPSO was launched on 28th April, 2006 with the cloud profiling radar system on the CloudSat satellite. They both fly in formation with three other satellites in the A-train constellation to enable an even greater understanding of the climate system from the broad array of sensors on these other spacecraft.

CALIOP measures attenuated aerosol backscatter profiles at 532 and 1064 nm, including parallel and perpendicular polarized components at 532 nm, with high variable horizontal and vertical resolution, for different atmospheric layers (i.e., aerosol, cloud and surface returns) (Kar et al., 2018; Vaughan et al., 2019). In order to implement the two-way transmittance method with CALIPSO data, the CALIPSO product used is the "Standard", with the Level 1 (L1) and Version 4.51 (V4.51), available from September 2022. This product has a horizontal (vertical) resolution depending on the altitude range, from 1/3 to 5 km

(30 to 300 m) and includes the mean, median and standard deviation of the total attenuated backscatter coefficient calibrated from non-ideal instruments effects associated with the polarization-sensitive at both wavelengths, along with their calibration constants.

## 2.3 Radiosoundings

Radiosondes are launched twice every day (at 00:00 and 12:00 UTC) by the Meteorological Service of Catalonia (Meteocat)

at a distance of less than 1 km from the MPL site. The radiosondes provide measurements of pressure, altitude, temperature, relative humidity, wind speed and direction. Only altitude, pressure and temperature profiles have been used in the present work.



## 3 The lidar two-way transmittance method

### 3.1 Literature review

In order to get reliable products of the optical properties of clouds and aerosols, different techniques are currently employed to invert elastic lidar signals. The solution of the inverse problem is not straightforward because there are two unknown parameters in the lidar equation: the backscatter and extinction coefficients. Therefore, over the years, this problem has been approached from several perspectives, such as (Fernald et al., 1972; Klett, 1981; Fernald, 1984; Klett, 1985), the two-way transmittance method (Evans, 1967; Charles et al., 1972; Platt, 1973; Young et al., 1995; Elouragini and Flamant, 1996; Del Guasta, 1998;

Chen et al., 2002; Platt et al., 2002; Cadet et al., 2005; Yorks et al., 2011; Córdoba-Jabonero et al., 2017) and others (Kovalev, 1993; Elouragini and Flamant, 1996).

In particular, the two-way transmittance method compares the lidar signals just below and above the cloud, assuming that the lidar signals correctly represent the scattering medium and that the zones below and above the cloud are aerosol/cloud-free or

molecular (Charles et al., 1972; Young et al., 1995; Del Guasta, 1998). On one hand, the main advantage of this method is that it does not require any a priori optical and/or microphysical hypotheses like the knowledge of the cloud lidar ratio, defined as the ratio of the cloud extinction to backscatter coefficients integrated over the cloud (Giannakaki et al., 2007). This parameter is not the same for all cirrus clouds and it depends on the ice crystal properties of cirrus clouds. Although, its value can be assumed to be in a range between 20-30 sr for ice clouds (Sassen and Comstock, 2001; Yorks et al., 2011; Lewis et al., 2016).

On the other hand, the major disadvantages of this method is that one has to make sure that the regions above and below the aerosol/cloud layer are molecular and so, it depends strongly on the aerosol-free quality of the normalization regions below and above the cirrus cloud. For this reason, it is necessary to select particle-free regions far enough from the cloud layer in order to normalise the signal, otherwise this method cannot be applied. Another disadvantage is that the retrievals are not accurate for very thin clouds (some studies suggest that the cloud optical depth must be upper than 0.1 (Cadet et al., 2005) or 0.05

(Chen et al., 2002)), for thick clouds because the lidar signal does not penetrate the whole cloud, for very noisy lidar signals or for small lidar signal values.

In spite of all these disadvantages, it is common to find this method combined with other ones, to make a first estimation of the cloud optical depth, due to its low computational cost. This first estimation of the cloud optical depth is usually used

as a constraint in other methods. For example, CALIPSO algorithm applies the transmittance method under certain situations. When a molecular region is found immediately above and below the cirrus cloud, the Hybrid Extinction Retrieval Algorithm (HERA) implemented with CALIPSO data uses the two-way transmittance method to obtain the cloud optical depth directly from the ratio of the mean attenuated scattering ratios, without multiple scattering correction (Young and Vaughan, 2009). It is also well known that Fernald method (Young et al., 1995) can be constrained by values of cloud transmittance determined by

the two-way transmittance method, (Elouragini and Flamant, 1996) combines the backward solution of the Klett method and the two-way transmittance method and (Cadet et al., 2005) shows the combination a method called particle integration method





(PI) with the two-way transmittance method to retrieve the optimal effective lidar ratio.

This method is based on the application of the lidar equation and the consideration of two reference points. For the calculation of the cloud optical depth, these points are placed above and below the cloud and the signal is normalized with the standard atmosphere, assuming molecular conditions at least in one of these regions. In this way, the power attenuation because of the cloud can be computed. There are many approaches of this method, applied to certain aerosol/cloud layers. The first works using this technique date back to the 1960s-70s, in which the authors calculated the transmittance of a smoke plume layer using lidar data (Evans, 1967; Charles et al., 1972). Over the years, the two-way transmittance method has been used to calculate the

cloud optical depth of cirrus clouds, considering different normalization regions or changing the extension of the normalization interval, the distance between the cirrus cloud and the normalization region, the time average applied to the lidar signal to reduce its noise, etc. For example, (Chen et al., 2002) normalizes the lidar signal on both sides of the cirrus cloud, particularly at the top and base of the cloud, that is in only two points of the vertical profile. On the contrary, (Cadet et al., 2005) considers only a normalization region below the cirrus cloud, extending from 0.7 km to 0.4 km below the cloud base and using 2-minutes

signals averaging. One last example, (Yorks et al., 2011) contemplates only a normalization region above the cloud, extended between 3-4 km below the aircraft and the top cloud height (typically between 13-16 km of altitude).

In this study, the two-way transmittance method has been applied to a case study, specifically a high-altitude cirrus cloud measured with the MPL and CALIOP at the same time, 11-02-2019 at 02:03:50 UTC in Barcelona. CALIPSO is at a distance

of 78 km from Barcelona station in that moment and to illustrate this cirrus case study, the CALIPSO signal has been analysed without any average, just a single shoot lidar profile.

### 3.2  For ground-based lidars

Following the notation of (Campbell et al., 2002), we call $NRB(z)$ the normalized relative backscatter or range-corrected signal at height $z$ and it can be written as:

$$NRB(z) = C\,\beta(z)\,T^2(z) = C\,[\beta_m(z) + \beta_p(z)]\,T_m^2(z)\,T_p^2(z) \tag{1}$$

where $C$ is the system calibration constant (for the method of solving $C$ see (Welton et al., 2001)) and $\beta$ and $T$ are the atmospheric backscatter and transmittance profiles, respectively. The molecular and particle contributions are denoted by $m$ and $p$ subscripts, respectively, as shown in the Eq. 1. In turn, the transmittance coefficient can be expressed as an exponential term as follow:

$$T_p(z) = \exp\left(-\int_0^z \eta(u)\,\alpha_p(u)\,du\right) \tag{2}$$

Where $\eta$ is the multiple scattering factor introduced by (Platt, 1973, 1979) and $\alpha_p$ is the particle extinction coefficient. We first calculate the attenuated molecular backscatter coefficient which is defined as:

$$\beta_m^{att}(z) = \beta_m(z)\,T_m^2(z) \tag{3}$$



Where $\beta_m$ and $\alpha_m$ are calculated using the equations of (U.S. Standard Atmosphere, 1976) with pressure and temperature

measurements from radiosondes. Then, we scale down the range corrected signal to the attenuated molecular backscatter

coefficient in the molecular region above the cloud, at height $z_t$. Where $z_t$ is the altitude corresponding to 0.2 km above the

cloud top height, also an input from MPLNET. The normalized NRB ($NRB_{nor}$) has the following expression:

$$NRB_{nor}(z) = \frac{\beta_m^{att}(z_t)}{NRB(z_t)} NRB(z) \tag{4}$$

In an aerosol-free atmosphere the vertical profiles of NRB and $NRB_{nor}$ would overlap. In practice, $\beta_m^{att}(z_t)$ and $NRB(z_t)$

are not calculated at a single point $z_t$. To compensate for the noise of NRB at high altitude, each quantity is calculated as the

mean value in an interval $[z_t, z_t+4.8]$ km above the cloud. The vertical extent of normalisation interval above the cloud has

been defined by performing different tests. Even though, its extension may vary, it is not critical, as the atmospheric region

above the cloud is assumed to be aerosol/cloud-free. The Eq. 4 can be extended as:

$$
\begin{aligned}
NRB_{nor}(z) &= \frac{\beta_m^{att}(z_t)}{NRB(z_t)} \, C \, \beta(z) \, T^2(z) \\
&= \frac{\beta_m(z_t) \, T_m^2(z_t)}{(\beta_m(z_t) + \beta_p(z_t)) \, T_m^2(z_t) \, T_p^2(z_t)} \, (\beta_m(z) + \beta_p(z)) \, T_m^2(z) \, T_p^2(z) \\
&= (\beta_m(z) + \beta_p(z)) \, T_m^2(z) \frac{T_p^2(z)}{T_p^2(z_t)},
\end{aligned}
\tag{5}
$$

Being $z_b$ the altitude corresponding to 0.2 km below the cloud bottom height, also an input from MPLNET. In practice, the

normalization of the attenuated molecular backscatter coefficient at $z_b$ is calculated as the ratio of mean quantities calculated in

an interval $[z_b, z_b-0.8]$ km below the cloud. This vertical extension of normalization is shallower than the normalization region

above the cloud because it is more likely to find a non-molecular atmospheric layer below the cirrus cloud. Its extension has

also been defined by performing different tests.

The ratio between the normalized range-square corrected signal coefficient in $z_t$ and the normalized attenuated molecular

backscatter coefficient in $z_b$ is:

$$
\begin{aligned}
\frac{NRB_{nor}(z_t)}{NRB_{nor}(z_b)} &= \frac{\beta_m(z_t) + \beta_p(z_t)}{\beta_m(z_b) + \beta_p(z_b)} \frac{T_m^2(z_t) \frac{T_p^2(z_t)}{T_p^2(z_t)}}{T_m^2(z_b) \frac{T_p^2(z_b)}{T_p^2(z_t)}} \\
&= \frac{\beta_m(z_t)}{\beta_m(z_b)} \frac{T_m^2(z_t)}{T_m^2(z_b) \frac{T_p^2(z_b)}{T_p^2(z_t)}} \\
&= \frac{\beta_m(z_t)}{\beta_m(z_b)} \frac{T_m^2(z_t)}{T_m^2(z_b)} \exp\left(-2 \int_{z_b}^{z_t} \eta(u) \, \alpha_p(u) \, du\right) \\
&= \frac{\beta_m^{att}(z_t)}{\beta_m^{att}(z_b)} \exp(-2 \, COD)
\end{aligned}
\tag{6}
$$



Where COD is the cloud optical depth defined as $COD = \int_{z_b}^{z_t} \eta(u)\, \alpha_p(u)\, du$. Finally, COD is calculated as:

$$COD = -\frac{1}{2} \ln \left[ \frac{\frac{NRB_{nor}(z_t)}{NRB_{nor}(z_b)}}{\frac{\beta_m^{att}(z_t)}{\beta_m^{att}(z_b)}} \right] = -\frac{1}{2} \ln \left[ \frac{\beta_m^{att}(z_b)}{NRB_{nor}(z_b)} \right] \tag{7}$$

This simplified expression is obtained because the normalized NRB matches the attenuated molecular backscatter coefficient at $z_t$, as can be derived from Eq. 4.

### 3.3 For spaceborne lidars

In order to follow with the same notation, we continue working with the NRB coefficient. Where the attenuated total backscatter coefficient $\beta^{att}(z)$ at height $z$, product provided by CALIPSO data (Hostetler et al., 2006), is defined as NRB coefficient with 240 the calibration constant one ($C = 1$, see Eq. 1).

$$NRB(z) = (\beta_m(z) + \beta_p(z))\, T_m^2(z)\, T_p^2(z) \tag{8}$$

Similarly to Section 3.2, we scale down the NRB to the normalized relative backscatter $NRB_{nor}$, resulting in the following expression:

$$
\begin{aligned}
NRB_{nor}(z) &= \frac{\beta_m^{att}(z_t)}{NRB(z_t)}\, NRB(z) \\
&= \frac{\beta_m(z_t)\, T_m^2(z_t)}{(\beta_m(z_t) + \beta_p(z_t))\, T_m^2(z_t)\, T_p^2(z_t)}\, (\beta_m(z) + \beta_p(z))\, T_m^2(z)\, T_p^2(z) \\
&= (\beta_m(z) + \beta_p(z))\, T_m^2(z)\, \frac{T_p^2(z)}{T_p^2(z_t)}
\end{aligned}
\tag{9}
$$

Being $z_t$ the altitude corresponding to 0.2 km above cloud top height and $z_b$ the altitude corresponding to 0.2 km below the cloud bottom height. In practice, the both normalizations are not calculated at a single point $z_t$, $z_b$, respectively. To compensate for the noise of NRB and $\beta_m^{att}$ at high altitude, each quantity is calculated as the mean value in a wide enough interval above/below the cloud, identically to the ground-base case, explained previously. The ratio between the normalized attenuated backscatter coefficient in $z_t$ and the normalized attenuated molecular backscatter coefficient in $z_b$ is:

$$
\begin{aligned}
\frac{NRB_{nor}(z_t)}{NRB_{nor}(z_b)} &= \frac{\beta_m(z_t)}{\beta_m(z_b)}\, \frac{T_m^2(z_t)}{T_m^2(z_b)}\, \exp\left( 2 \int_{z_b}^{z_t} \eta(u)\, \alpha_p(u)\, du \right) \\
&= \frac{\beta_m^{att}(z_t)}{\beta_m^{att}(z_b)}\, \exp\left( 2\, COD \right)
\end{aligned}
\tag{10}
$$


Finally, COD is calculated as:

$$COD = \frac{1}{2} \ln \left[ \frac{\frac{NRB_{nor}(z_t)}{NRB_{nor}(z_b)}}{\frac{\beta_m^{att}(z_t)}{\beta_m^{att}(z_b)}} \right] = \frac{1}{2} \ln \left[ \frac{\beta_m^{att}(z_b)}{NRB_{nor}(z_b)} \right] \tag{11}$$





### 3.4 Unified formulation

Once the mathematical developments for the application of the two-way transmittance method for ground-based lidars (see

Section 3.2) and spaceborne lidars (see Section 3.3) are shown, an example case of cirrus cloud is analysed, as shown in

Fig. 1a, 1b, respectively.

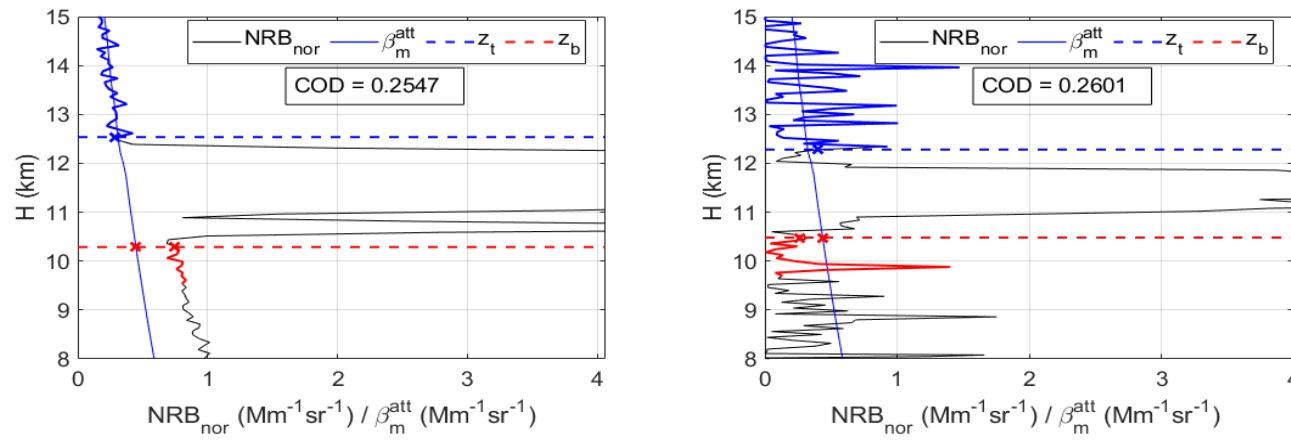

**Figure 1.** Application of the two-way transmittance method for (left) MPLNET and (right) CALIPSO data, for the case 11/02/2019 at 02:03:50 UTC. The height $z_b(z_t)$ is the altitude corresponding to 0.2 km above (below) cloud top (base) height.

Fig. 1a, 1b show the application of the two-way transmittance method to a cirrus cloud detected in Barcelona, with the

MPL and CALIOP lidar, respectively, at the same time. It is discerned that the cirrus cloud measured with ground-based lidar

has a lower base than the cirrus cloud detected with spaceborne lidar. This fact is to be expected as the satellite is found at a

distance of 78 km from Barcelona station. Consequently, the two lidar systems measure different parts of the same cirrus cloud

or simply two different cirrus clouds in close proximity. In spite of this, applying the two-way transmittance method gives

similar COD, being 0.2547 for the cirrus cloud detected by MPL lidar and 0.2601 for the cirrus cloud measured by CALIOP

lidar. Thus, the results obtained from the two-way transmittance method for ground-based and spaceborne lidars are equivalent.

Returning to the mathematical development, after COD calculation, we can estimate the Lidar Ratio of the whole cloud

$(LR_{cloud})$ using the following equation (Chen et al., 2002):

$$LR_{cloud} = \frac{\int_{cbh}^{cth} \eta(x)\, \alpha_p(x)\, dx}{\int_{cbh}^{cth} \beta_p(x)\, dx} \tag{12}$$

The particle backscatter is solved out from Eq. 5 being:

$$\beta_p(z) = \frac{NRB_{nor}(z)}{T_m^2(z)} \left( \frac{T_p^2(z)}{T_p^2(z_t)} \right)^{-1} - \beta_m(z) \tag{13}$$





Where $\left(\frac{T_p^2(z)}{T_p^2(z_t)}\right)^{-1} = \exp\left(-2\int_z^{z_t} \eta(z)\,\alpha_p(z)\,dz\right)$. With the aim to calculate the $\beta_p(z)$, in the first iteration ($k=1$) it is assumed that the extinction coefficient profile in the whole cloud is constant as:

$$\alpha_{p,1} = \frac{COD}{CT} \tag{14}$$

Where COD is calculated with the Eq. 7 or Eq. 11 and CT is the cloud thickness, which is the difference between cloud top height and cloud base height, provided by MPLNET. After this first step, a new extinction coefficient profile is calculated as:

$$\alpha_{p,k+1}(z) = \eta(z)\,LR_{cloud,k}\,\beta_{p,k}(z) \tag{15}$$

Afterwards, the new extinction coefficient profile, $\alpha_{p,k+1}$, is used to calculate the next particle backscatter profile ($\beta_{p,k+1}$). This process is continued until successive values of the $LR_{cloud}$ integral and the previous one differ negligibly, in other words $|LR_{cloud,k+1} - LR_{cloud,k}| < 1$ sr.

In order to study the optical characteristics of the cirrus clouds, we calculate the linear cloud depolarization ratio LCDR,

that is defined as the ratio of the perpendicular and parallel lidar signals in the cloud (Chen et al., 2002). This parameter is not directly provided by CLD MPLNET product. Instead, the volume depolarization ratio (VDR) is given by CLD MPLNET product.

$$LCDR(z) = \frac{P_\perp(z)}{P_\parallel(z)} \tag{16}$$

The vertical profile of the linear cloud depolarization ratio can be calculated by means of the following expression (Freuden-

thaler et al., 2009)

$$LCDR(z) = \frac{[1+MDR]\,VDR(z)\,R(z) - [1+VDR(z)]\,MDR}{[1+MDR]\,R(z) - [1+VDR(z)]} \tag{17}$$

Being $MDR$ the molecular depolarization ratio and $R$ the backscatter ratio, that is defined as

$$R(z) = \frac{\beta_m(z) + \beta_p(z)}{\beta_m(z)} \tag{18}$$

According to (Behrendt and Nakamura, 2002), $MDR$ has a value of 0.00363. Once the vertical profile of the linear cloud

depolarization rate has been obtained, the coefficient associated to the whole cloud is determined as the average of a half-cloud vertical profile, centred at the maximum peak, shown in Fig. 2.





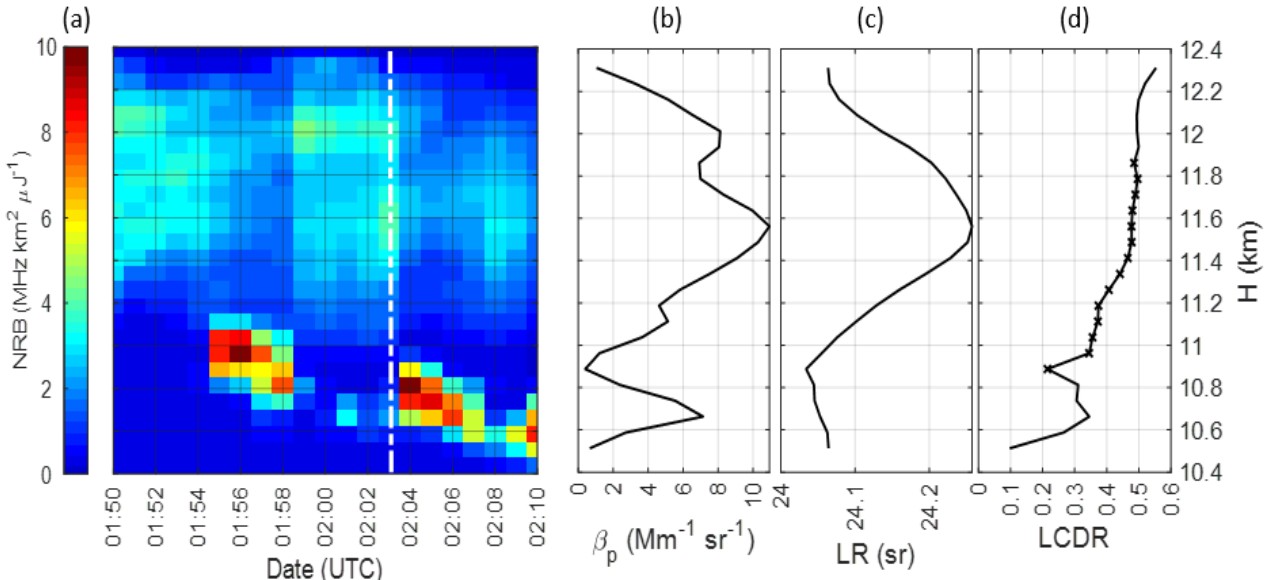

**Figure 2.** Characterization of the cirrus cloud 11/02/2019 at 02:03 UTC measured in Barcelona. (a) Minute evolution and vertical profiles of (b) the particle backscatter, (c) the lidar ratio and (d) the linear cloud depolarization ratio at 532 nm. The white line indicates the exact cirrus detection time and the black crosses refer to the cirrus zone where it is averaged to obtain the linear cloud depolarization ratio coefficient.

Fig. 2 shows a quick look of the NRB within the cirrus cloud, together with the associated vertical profiles of particle backscatter, lidar ratio and linear cloud depolarization ratio calculated with the two-way transmittance method for the case study of 11/02/2019 at 02:03 UTC. It is observed that the peaks of the particle backscatter coefficient correspond to the areas

of the cloud with the highest NRB values. The LR curve also presents a similar shape to that of the particle backscatter coefficient but smoother, varying its value only a couple of tenths over the entire vertical profile of the cloud. On the contrary, the LCDR has a flatter shape throughout its vertical profile and has some oscillations in the lowest cloud layer. To avoid these more irregular areas, the average of a half-cloud vertical profile, centred at the maximum peak is calculated to obtain a representative value for the whole cirrus.

**4   Five years of cirrus retrievals**

Even though, there is no a widely criterion accepted for the identification of cirrus clouds, the most common definition of cirrus clouds is that they must be composed mainly of ice crystals. This is because their geometrical and optical properties vary with the latitude, as illustrated by the different cirrus identification criteria in Table 1, established in the literature. In this study, the criteria adopted for the identification of high-altitude cirrus clouds in Barcelona is based on two conditions. (1) The

temperature at the cloud top height must be lower than -37ºC (Sassen and Campbell, 2001; Campbell et al., 2015) and (2) the cloud base height must be upper than 7 km, to ensure cirrus clouds, as opposed to other types of clouds.





| Measurement site | Location | Criteria | Reference |
|---|---|---|---|
| Fairbanks, Alaska | 64.86ºN, 147.85ºW; 300 m a.s.l. | $T_{top} < -37^oC$ <br> $T_{top} > -75^oC$ | (Campbell et al., 2018) |
| Lille, France | 50.60ºN, 3.14ºE; 21 m a.s.l. | $T_{base} < -25^oC$ | (Rita et al., 2016) |
| Barcelona, Spain | 41.38º N, 2.11º E; 115 m a.s.l. | $CBH > 7\,km$ <br> $T_{top} < \text{-37°C}$ | **This study** |
| Thessaloniki, Greece | 40.6ºN, 22.9ºE; 250 m a.s.l. | $T_{mid} < -38^oC$ | (Giannakaki et al., 2007) |
| Greenbelt, Maryland | 38.99ºN, 76.84ºW; 50 m a.s.l. | $T_{top} < -37^oC$ | (Campbell et al., 2015) |
| Hulule, India | 4.11ºN, 73.31ºE; 3 m a.s.l. | $CBH > 9\,km$ | (Seifert et al., 2007) |
| Bangor, Maine <br> Warner-Robbins, Georgia <br> Houston, Texas <br> Honolulu, Hawaii <br> San José, Costa Rica | 44.82ºN, 68.83ºW; 36 m a.s.l. <br> 32.64ºN, 83.59ºW; 93 m a.s.l. <br> 29.60ºN, 95.16ºW; 24 m a.s.l. <br> 21.32ºN, 157.92ºW; 6 m a.s.l. <br> 9.99ºN, 84.21ºW; 1172 m a.s.l. | $CH > 8\,km$ <br> $T_{mid} < -20^oC$ <br> $VDR > 0.27$ | (Yorks et al., 2011) |
| Kuopio, Finland <br> Gwal Pahari, India <br> Elandsfontein, South Africa | 62.74ºN, 27.54ºE; 190 m a.s.l. <br> 28.43ºN, 77.15ºE; 243 m a.s.l. <br> 26.25ºS, 29.43ºE; 1745 m a.s.l. | $CBH > 6\,km$ <br> $T_{top} < -38^oC$ <br> $T_{base} < -27^oC$ | (Voudouri et al., 2020) |
| Santa Cruz de Tenerife, Spain <br> Sao Paulo, Brazil <br> Belgrano, Argentina | 28.5ºN, 16.3ºW; 92 m a.s.l. <br> 23.6ºS, 46.8ºW; 760 m a.s.l. <br> 78ºS, 35ºW; 18 m a.s.l. | $T_{top} < -38^oC$ | (Córdoba-Jabonero et al., 2017) |

**Table 1.** Summary of criteria for cirrus clouds identification, reported in literature. Where $T_{mid}$, $T_{top}$ and $T_{base}$ are the temperatures at cloud mid, top and base heights, respectively; $CH$ and $CBH$ are the mid-cloud and the cloud base heights, respectively and $VDR$ is the volume depolarization ratio.

Table 1 shows that most cirrus identification criteria are based on the height or temperature of the cirrus clouds at the base, top or middle altitudes. Even many of them set criteria on both sides of the cloud, to ensure that it does not contain liquid water. Although used by some authors, e.g. (Yorks et al., 2011), the depolarization criterion is not widely used because of the low LCDR values of horizontally oriented ice (HOI) crystals (Hu et al., 2009) which might lead to discard erroneously cirrus clouds made of such ice crystals.

In this study, cirrus clouds are considered as the highest clouds in a vertical profile. In order to assure Rayleigh regions both above and below the cirrus cloud to be analysed, if there is another cloud lower, less than 1 km away, the two clouds are merged and treated as one cloud layer. In all cases, it has also been imposed that the lidar signal is not extinguished behind the cloud. After the classification of cirrus scenes, the two-way transmittance method has been applied to our database. Cases with LR





higher than 100 sr (generally cases with very noisy lidar signal) were discarded. A statistical analysis will be presented and discussed in Sections 4.1, 4.2.

## 4.1 Cirrus geometrical properties

After having carried out the identification of 367 high-altitude cirrus clouds, measured in Barcelona, through MPLNET products and radiosonde data from November 2018 to September 2022 (only at 00 and 12 UTC, when radiosondes are available), the two-way transmittance method has been applied successfully to 203 of them, i.e. to 55% of all cases. Note that 39% percent of the 203 high-altitude cirrus cases have another cloud below the cirrus cloud. The elimination of some cases has been carried out on the basis of the noise associated to the lidar signal and/or the possibility to guarantee a cloud/aerosol-free zone

both above and below the cirrus cloud. In this section, the geometrical and optical properties of high-altitude cirrus scenes are presented and discussed.

The cirrus occurrence in Barcelona, together with the monthly distribution of cirrus scenes classified as follows: sub-visible (SVC; COD < 0.03), visible (VC; 0.03 < COD < 0.3) and opaque (COD > 0.3) cirrus cloud according to (Sassen and Cho,

1992) criteria, have been analysed, are shown in Fig. 3a and 3b, respectively.

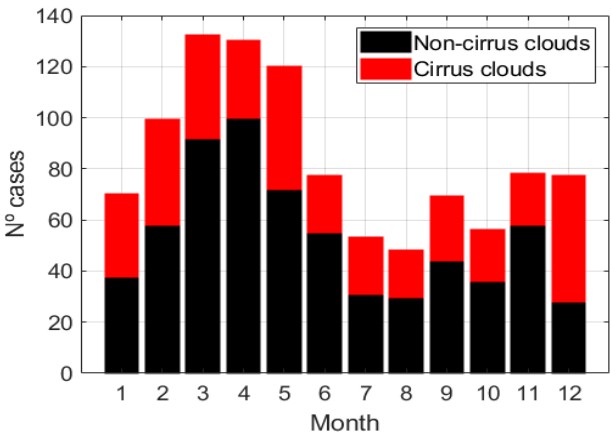
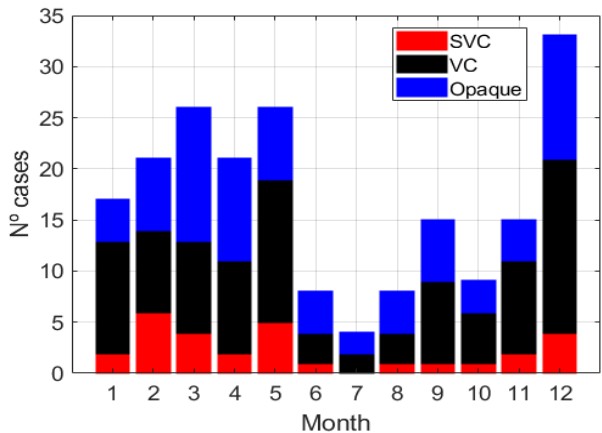

**Figure 3.** Histogram of the number of (left) cases of non-cirrus and cirrus clouds and (right) all cirrus clouds, for which the two-way transmittance method have been applied, classified as sub-visible (red), visible (black) and opaque (blue), for each month from 2018 to 2022 in Barcelona.

The cloud detection has been performed with MPLNET CLD product. When in a 1-min vertical profile there was a valid cloud base and cloud top value, it was counted as a cloud case. In Fig. 3a, it is further noted that the percentage of cirrus cases is not negligible, being a 36% of 1019 cloud cases at 00 and 12 UTC, during the five years. Of this percentage, 39% of cirrus cases have another cirrus below them, specifically at a distance of lower than 1 km, and both clouds are merged and





considered as one cloud layer. Cirrus cloud maximum occurrence is in spring, due to the fact that this is a time of great synoptic atmospheric instability in the Iberian Peninsula. In the mid-latitude regions, the formation process of cirrus clouds is linked to the deep convective outflow (Li et al., 2005; Fu et al., 2006; Jin et al., 2006), the synoptic scale lifting of air leading to the ice nucleation at low temperatures (Das et al., 2010), and the cooling associated to the wave activity in the upper troposphere (Spichtinger et al., 2003). This phenomenon has also been observed in other studies as (Giannakaki et al., 2007; Rita et al., 2016), where the highest frequency of mid-latitude cirrus is in autumn and spring.

Fig. 3b shows that the most abundant cirrus type is visible cirrus (48%), followed by opaque (38%) and a minority are sub-visible cirrus (14%). The monthly distribution of sub-visible cirrus clouds does not vary considerably, remaining the category with the lowest occurrence over the year. In contrast, the distribution of visible and opaque cirrus varies slightly. It can be said that in the warmer seasons, opaque cirrus are more frequent than visible cirrus. As shown in Table 3, these observations vary considerably depending on the latitude. The frequency of cirrus detection seems to be highly variable, a more extended database is needed to state a tendency. It is also observed that at latitudes close to the one of this study (41.38ºN, Barcelona), the occurrence of visible cirrus clouds predominates. In general, it could be said that the occurrence of each cirrus depends on the weather pattern of each site.

The probability distribution of cloud base and top heights and mid-cloud temperature of cirrus clouds are shown in Fig. 4.

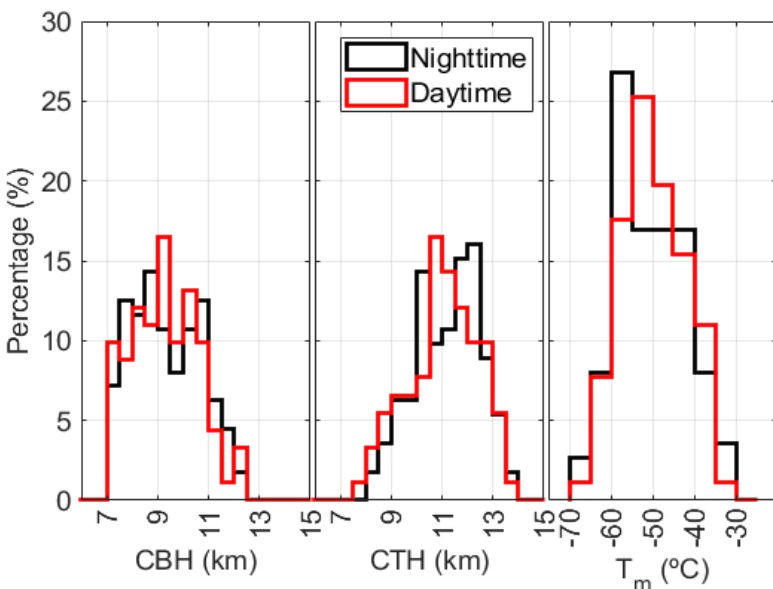

**Figure 4.** Probability distribution of (left) cloud base height; (center) cloud top heights; (right) mid-cloud temperature, of cirrus clouds at nighttime (00 UTC; black) and daytime (12 UTC; red) from 2018 to 2022 in Barcelona.





Ground-based elastic lidars are very sensitive to the solar background noise. For that reason the nighttime and daytime contributions have been separated. In Fig. 4, it can be seen that the probability distribution of cloud base and top altitudes range respectively from 7 to 12.5 km and 7.5 to 14 km, with peaks of occurrence at 9 and 10 km, respectively. Daytime and nighttime

results are very similar. The mid-cloud temperature ranges between -30 to -70 ℃ and has a maximum around -55 (-50) ℃ during nighttime (daytime). These results fit well with the literature, specifically studies carried out at similar latitudes like (Rita et al., 2016) who obtained distribution of cloud base (top) altitudes ranging from 5 to 13 (5 to 14) km, with a mean value of 8 (11) km. The mid-cloud temperature ranges between -30 to -80 ℃ and has two maximum peaks of occurrence around -45 and -55 ℃. Another example is (Campbell et al., 2016) who got cloud top altitude between 6 and 16 km, with a mean top at

11 km and cloud top temperature between -35 and -75 ℃, with two maximum peaks of occurrence around -50 and -60 ℃, for daytime cirrus clouds.

In order to better analyse the geometrical properties of the 203 high-altitude cirrus cases measured in Barcelona, during the years 2018 to 2022, mean values and standard deviations have been calculated and are shown in Table 2.

|  | CBH (km) | CTH (km) | CH (km) | CT (km) | $T_{base}$ (℃) | $T_{top}$ (℃) | $T_m$ (℃) | Nº cirrus (%) |
|---|---|---|---|---|---|---|---|---|
| **Nighttime** | 9.4±1.4 | 11.2±1.3 | 10.3±1.2 | 1.9±1.1 | -43.6±10.3 | -57.9±8.7 | -50.7±8.4 | 112 (55) |
| **Daytime** | 9.3±1.3 | 11.0±1.4 | 10.2±1.2 | 1.7±1.2 | -44.1±9.5 | -56.8±8.6 | -50.5±8.0 | 91 (45) |

**Table 2.** Average and standard deviation values of geometrical properties of cirrus clouds at nighttime (00 UTC) and daytime (12 UTC) from 2018 to 2022 in Barcelona. Where CH is mid-cloud height, N the number of cirrus clouds ans (%) its percentage with respect to the total number of cirrus clouds to which the two-way transmittance method has been applied successfully.

Table 2 shows that the number of cirrus clouds analysed at nighttime is similar to that at daytime, making the results comparable. Furthermore, the percentage of cirrus cases at nighttime and daytime to which the two-way transmittance method has been successfully applied, compared to those identified at these hours does not differ considerably according to the hour of day, being 51% for cirrus cases at nighttime and 62% at daytime. As a consequence, it can be stated that solar background radiation does not affect the efficiency of the two-way transmittance method.


It can be also observed that the cloud base and top heights together with the cloud thickness are higher at nighttime than at daytime. Consequently, mid-cloud temperature is lower at nighttime than at daytime. These differences are negligible due to their values are lower than their standard deviation, being a similar result to that obtained in (Rita et al., 2016). It could be said that the macrophysical properties of cirrus clouds do not vary with the time of the day. The distribution of high-altitude

cirrus thickness is in the range from 1 to 5 km, with a mean value of 1.9 km, which means that clouds with a large vertical development, characteristic of other cloud types than cirrus clouds have been correctly discarded.





This results fit well with the literature, in which diverse studies such as (Sassen and Campbell, 2001) shows that the cloud base height is 8.79 km and (Rita et al., 2016) distinguishing between daytime and nighttime measurements get values of 8.92±1.65 km and 8.91±1.60 km, respectively. These values are slightly lower than those obtained in this study carried out in Barcelona, but belong to the distribution shown in Fig 4. Regarding the cloud top height, (Sassen and Campbell, 2001) obtains a very similar value to that of this study, being 11.2 km, together with (Campbell et al., 2015) that gets 11.15 km and (Rita et al., 2016) shows lower values of cloud top heights of 10.46±1.59 km and 10.62±1.50 km, for daytime and nighttime measurements. Instead of analysing cloud base and top heights, other studies like (Dowling and Radke, 1990), examines typical values of cirrus cloud altitude between 4 to 20 km, with the peak of occurrence value of 9 km, being lower than the value calculated in this study.

With respect to the thickness of cirrus clouds, (Dowling and Radke, 1990) shows values ranging between 0.1 to 8 km, with the peak of most occurrence value of the distribution of 1.5 km, (Sassen and Campbell, 2001) gets a averaged value of 1.81 km and (Rita et al., 2016) shows values of 1.54±0.91 km and 1.71±0.93 km, for daytime and nighttime measurements, respectively. These values are also lower than those obtained in this study, indicating that the cirrus measured in this study are thicker. This fact could be due to the fact that 39% of the cirrus cases have another cirrus below them, specifically at a distance of less than 1 km, and both are considered as one.

Continuing with the analysis of the physical and geometrical properties of cirrus clouds, there are studies such as (Sassen and Campbell, 2001) which show a temperature value at cloud base of -34.4 ℃, being a value considerably higher than the value obtained in this study. Regarding the temperature at cloud top, (Sassen and Campbell, 2001) shows a value is slightly higher than the value of our study, which is -53.9 ℃. To complete the cirrus temperature analysis, (Campbell et al., 2015) gets a mid-cloud temperature of -58.47 ℃ and making the difference between daytime and nighttime measurements, (Rita et al., 2016) shows values of -49±10 ℃ and -50±9 ℃, respectively. Where the values obtained by (Rita et al., 2016) are really much closer to this study than that of (Campbell et al., 2015).

## 4.2 Cirrus optical properties

Probability distributions of the following optical properties: cloud optical depth, lidar ratio and linear cloud depolarization ratio, calculated using the two-way transmittance method (see Section 3), have also been determined for all the cirrus scenes and are shown in Fig 5.





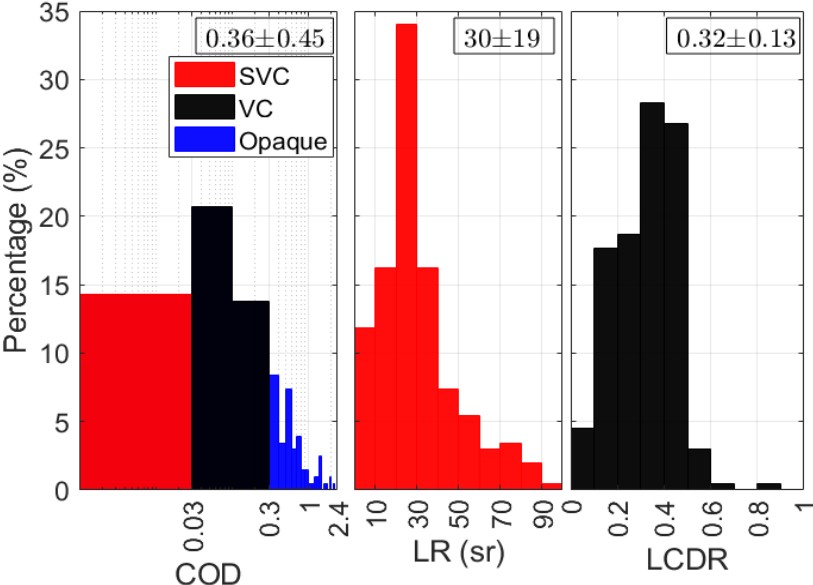

**Figure 5.** Probability distribution of (left) cloud optical depth, (center) lidar ratio and (right) linear cloud depolarization ratio, calculated using the two-way transmittance method, from 2018 to 2022 in Barcelona. The rectangles in the upper right-hand corners show average values and standard deviations of the distributions. The left figure has a logarithmic grid to show the sub-visible cloud group.

In Fig 5 one observes that the presence of visible cirrus clouds dominate this study, with an occurrence of 48%, followed by opaque cirrus clouds with a percentage of 38% and lastly, the sub-visible cirrus clouds with a 14% of occurrence. Many studies point to the fact that the high-altitude cirrus clouds have an optical depth usually lower than 0.3 (Reichardt, 1999; Sassen and Campbell, 2001; Lee et al., 2009), being the case in 62% of the cases analysed in this study. However, this pa-
rameter can vary from 0.003 to 3 (Sassen and Comstock, 2001). The mean value of the distribution is 0.36, being a value much larger than 0.1 because cirrus clouds with much larger COD alter the mean, as reflected in the standard deviation of the distribution. The lidar ratio varies mostly between 20 to 30 sr (33%), in agreement with the literature (Sassen and Comstock, 2001; Yorks et al., 2011; Josset et al., 2012; Garnier et al., 2015; Córdoba-Jabonero et al., 2017), with an average value of 30 sr. The linear cloud depolarization ratio is tipically between 0.3-0.5 (54%), with an average of 0.32, which is in agreement with
(Sassen, 2005; Giannakaki et al., 2007; Kim et al., 2018; Hu et al., 2021). The lowest values of the linear cloud depolarization ratio may be due to an ice crystals horizontal orientation tendency or to signal noise (Hu et al., 2009), since in 39% percent of the cases, another cloud was detected below the cirrus cloud.

**Table 3.** Summary of characteristics of cirrus clouds of ground-based lidar observations, reported in literature. The optical properties have
been calculated at 532 nm. Where N is the number of cirrus clouds identified and (%) its percentage with respect to the total number of clouds. The occurrence of SVC, VC and opaque cirrus clouds are made on the number of cirrus N. (a) $T_m$ values have been manually calculated from values of temperature at cloud and top heights, shown in the paper. (b) The geometrical properties show are from an annual average and





the optical properties are obtained by the two-way transmittance method applying a multiple scattering correction. (c) The optical properties are calculated at 355 nm.





| Measurement site (Time period) | Location | N° cirrus (%) | Occurrence (%) | | | CBH (km) | CTH (km) | $T_m$ (°C) | CT (km) | COD | LR (sr) | Reference |
|---|---|---|---|---|---|---|---|---|---|---|---|---|
| | | | SVC | VC | Opaque | | | | | | | |
| Kuopio (2012-2016) | 62.74°N, 27.54°E 190 m a.s.l. | 213 | 3 | 71 | 26 | 8.6±1.1 | 9.8±1.1 | −50±10 | 1.2±0.7 | 0.25±0.2 | 31±7 | (Voudouri et al., 2020)[a] |
| Haute of Provence (1996-2007) | 43.9°N, 5.7°E 679 m a.s.l. | 1850 (37) | 38 | | | 9.3±1.8 | 10.9±1.7 | | 1.6±1.1 | | | (Hoareau et al., 2013) |
| Rome (2007-2010) | 41.8°N, 12.6°E 107 m a.s.l. | 107 (30) | 10 | 49 | 41 | | | | | 0.37±0.18 | 31±15 | (Dionisi et al., 2013) |
| Barcelona (2018-2022) | 41.38°N, 2.11°E 115 m a.s.l. | 367 (36) | 14 | 48 | 38 | 9.3±1.3 | 11.1±1.3 | −51±8 | 1.8±1.1 | 0.36±0.45 | 30±19 | **This study** |
| Thessaloniki (2000-2006) | 40.6°N, 22.9°E 250 m a.s.l. | 53 | 3 | 57 | 40 | 9.0±1.1 | 11.7±0.9 | −51±6 | 2.7±0.9 | 0.34±0.32 | 29±24 | (Giannakaki et al., 2007)[b] |
| Naqu (Jul-Aug 2011) | 31.5°N, 92.1°E 4508 m a.s.l. | (15) | 16 | 34 | 50 | 13.7±2.0 | 15.6±1.6 | | 1.7 | 0.33±0.29 | 28±15 | (He et al., 2013) |
| Gwal Pahari (2008-2009) | 28.43°N, 77.15°E 243 m a.s.l. | 11 | 0 | 20 | 80 | 9.0±1.6 | 10.6±1.8 | −39±5 | 1.5±0.7 | 0.45±0.3 | 28±22 | (Voudouri et al., 2020)[a] |
| Gadanki (1998-2013) | 13.5°N, 79.2°E 370 m a.s.l. | | 52 | 36 | 11 | 13.0±2.2 | 15.3±2.0 | −65±12 | 2.3±1.3 | | | (Pandit et al., 2015) |
| Hulule (1999-2000) | 4.11°N, 73.31°E 3 m a.s.l. | 179 (43) | 8 | 52 | 40 | 11.9±1.6 | 13.7±1.4 | −58±11 | 1.8±1.0 | 0.28±0.29 | 32±10 | (Seifert et al., 2007) |
| Amazonia (2011-2012) | 2.89°S, 59.97°W 60 m a.s.l. | (74) | 42 | 38 | 20 | 12.9±2.2 | 14.3±1.9 | | 1.4±1.1 | 0.25±0.46 | 23±8 | (Gouveia et al., 2017)[c] |
| Elandsfontein (2009-2011) | 26.25°S, 29.43°E 1745 m a.s.l. | 64 | 2 | 61 | 37 | 9.2±0.8 | 10.8±0.9 | −40±6 | 1.6±0.7 | 0.3±0.3 | 25±6 | (Voudouri et al., 2020)[a] |
| Buenos Aires (2010-2011) | 34.6°S, 58.5°W 1572 m a.s.l. | | 0 | 68 | 32 | 8.3 | 11.8 | −65±4 | 3.0±0.9 | 0.26±0.11 | 33±17 | (Lakkis et al., 2015) |





Table 3 shows that the averages of cloud base (top) height range from 8 (10) to 14 (16) km, approaching the tropopause in some cases. Mid-latitude cirrus clouds are not found at altitudes below 7 km, so the criterion previously established for cirrus identification is correct. It also appears that the cirrus base and top height distributions are not dependent on latitude, but rather on cirrus type. Thinner cirrus like SVC are usually found at higher altitudes than opaque cirrus. This relation will be studied in the next subsection (see Subsection 4.3). Mid-cloud temperatures are in the range of -40 to -65 ºC and the cloud thickness between 1 and 3 km. The optical properties of the clouds are very

similar to those obtained at similar latitudes and the lidar ratio seems to have a generally increasing trend towards the poles.

## 4.3   Cirrus classification

A complementary analysis is carried out in this section, classifying the cirrus according to the criteria of (Sassen and Cho, 1992). For this purpose, the averages and standard deviations of the geometrical and optical properties of the cirrus clouds are calculated, as shown in Table 4.

| Types | CBH (km) | CTH (km) | $T_m$ (ºC) | CT (km) | COD | LCDR | LR (sr) | Nº cases |
|-------|----------|----------|------------|---------|-----|------|---------|----------|
| SVC | 10.2±1.2 | 11.1±1.4 | -55±7 | 0.9±0.6 | 0.02±0.01 | 0.27±0.17 | 17±19 | 29 |
| VC | 9.7±1.3 | 11.3±1.3 | -53±8 | 1.6±1.0 | 0.14±0.09 | 0.33±0.12 | 29±17 | 98 |
| Opaque | 8.6±1.0 | 11.0±1.4 | -47±7 | 2.4±1.2 | 0.78±0.5 | 0.33±0.13 | 36±18 | 76 |

**Table 4.** Average and standard deviation of optical properties of cirrus clouds classified with (Sassen and Cho, 1992) criteria from 2018 to 2022 in Barcelona.

       In Table 4, it can be seen that the cloud top height do not vary considerably depending on the type of cloud. The cirrus clouds might reach to/near the tropopause, since the average tropopause height calculated with radiosondes (World Meteorological Organization, 1957) on the days of cirrus scenes analysed is 11± 1 km. In contrast, the other geometrical, thermal and optical properties do vary with cloud type.

For example, subvisible clouds are the highest, coldest and thinnest clouds. Also, their thickness is 0.9 km less than the average thickness calculated with the whole cirrus dataset and its temperature is 4ºC colder than the mean temperature. These results are consistent with other studies of SVC cirrus from space-borne lidar observations (Martins et al., 2011). Their COD is within the value selected to make this classification and the LR is lower than 30 sr, and their mean linear cloud depolarization ratio is 0.27, being the lowest value of all categories. Visible cirrus clouds are the most predominant type in this study. Their geometrical properties are very similar to those of the whole cirrus

dataset, but the average of the optical properties are slightly lower. Opaque cirrus clouds have the highest value of LR, which may be due to the fact that these clouds contain the greatest richness and variety of ice crystals. On the other hand, opaque clouds contribute the most to the total radiative forcing (Kienast-Sjögren et al., 2016), being the lowest, warmest and thickest clouds in the whole cirrus dataset. Also, their thickness is 0.6 km higher than the average thickness calculated with the whole cirrus dataset and its temperature is 4ºC warmer than the mean temperature.

## 4.4   Cirrus correlation

In this section the correlations between the different cirrus products obtained with the two-way transmittance method, radiosonde and MPLNET data are analysed. First, the linear correlations between the temperature and height of the cirrus base and the lidar ratio with





the cloud optical depth are analysed, as shown in Fig 6. The cirrus clouds have been classified according to the (Sassen and Cho, 1992) criteria.

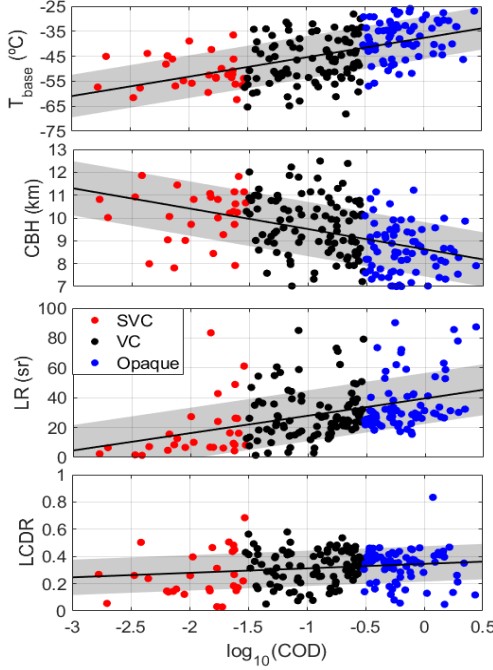

**Figure 6.** Logarithmic dependence of the cloud optical depth with (a) cloud base temperature, (b) cloud base height, (c) lidar ratio and (d) linear cloud depolarization ratio, for cirrus cases from 2018 to 2022 in Barcelona. The black line is the linear regression that has been calculated between the variables and the grey shading is the 95% confidence limit of the linear regression. The $R^2$ coefficients are (a) 0.26, (b) 0.19, (c) 0.17 and (d) 0.03.

On the one hand, Fig 6 shows a weak positive linear dependence between the logarithm of the cloud optical depth and the cloud base temperature and contrary to this, a negative tendency with the cloud base height. This means that as the cloud base temperature increases, the cloud base height decreases, being characteristic of the troposphere. As the cloud base height is lower, it is observed that the cloud optical depth increases. This could be due to the fact that as an air mass loaded with water vapour ascends vertically, the water vapour gradually condenses. Thus, clouds located at higher altitudes are formed from air masses with a lower water vapour content and, therefore, both their

geometric and optical thickness are smaller. An example of this phenomenon are sub-visible cirrus clouds, which are the highest, coldest and thinnest clouds and have the lowest COD values.

On the other hand, the lidar ratio increases with increasing cloud optical depth, a fact that has been observed in other researches (Chen et al., 2002; Dionisi et al., 2013). It is known that the lidar ratio indicates the complexity of ice crystal shape and aspect ratio (Sassen,

1978; Takano and Liou, 1995). When the complexity of ice crystal shape and diversity increases, the lidar ratio also increases (Seifert et al., 2007). Having clouds with a higher COD implies that the cloud base height is at lower levels, as mentioned above, so that there are larger and more irregular ice particles, due to collisions and turbulence, increasing the lidar ratio (He et al., 2013). This phenomenon is also seen,





for example, in sub-visible cirrus clouds, which generally have the lowest LR values.

Likewise, the linear cloud depolarization ratio has a slightly positive tendency with the cloud optical depth, being opposed to the research of (Chen et al., 2002). This tendency might be due to the fact that as the COD increases, the number of ice crystals increases and, as a consequence, the randomly aggregation of ice crystals within the cloud occurs more frequently. As the ice crystals increase in size, they become rougher and consequently, depolarization increases (Yang et al., 2000).

To conclude this section, Fig 7 shows the relationship between the linear cloud depolarization ratio and the lidar ratio calculated with the two-way transmittance method, classifying the cirrus clouds according to (Sassen and Cho, 1992) criteria.

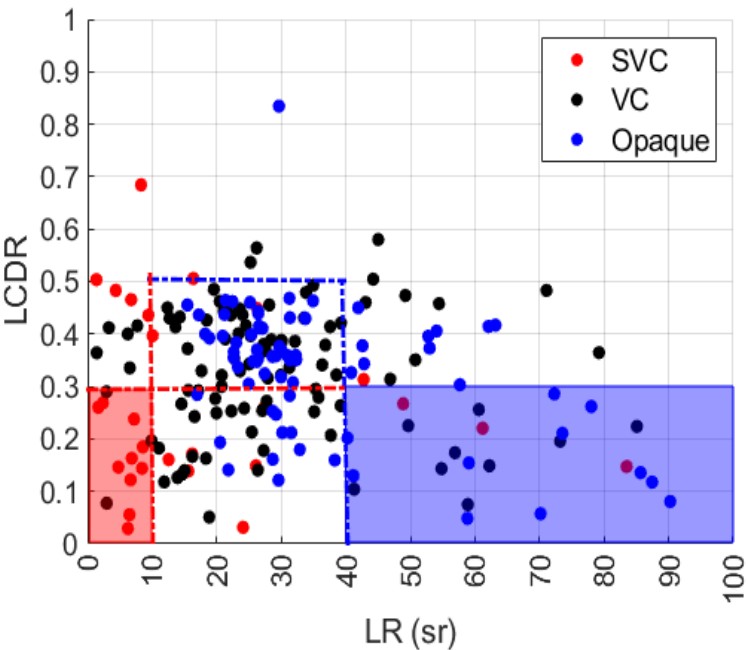

**Figure 7.** Dependence of the linear cloud depolarization ratio with the lidar ratio and and the cloud optical depth, for cirrus cases from 2018 to 2022 in Barcelona. The rectangles indicate the areas where the LCDR is lower than 0.3 and for lidar ratio values out of the known range for cirrus clouds.

In Fig 7 it can be seen that there is no linear dependence between the linear cloud depolarization ratio and the lidar ratio of the cirrus clouds. At the same time, all three types of cirrus clouds are found in all sectors of the scatter plot, with more or less frequency. On the one hand, in the red sector, sub-visible cirrus clouds clearly dominate, being the highest, coldest and thinnest clouds. Because of their geometrical and
optical properties, the possibility that these cirrus clouds could contain liquid water is ruled out. On the other hand, visible and opaque cirrus clouds control the blue sector. The percentage of cirrus found in this area is 12%. As one cloud type does not predominate, the geometrical properties of this subgroup have been analysed, showing a mid-cloud height of $11.9 \pm 1.4$ km and a mid-cloud temperature of $-63.7 \pm 6.5$ ºC, making the presence of aqueous content in these cirrus clouds impossible.


## 5 Conclusions

In this study, the cirrus geometrical and optical properties of 5 years of continuous ground-base lidar measurements with the Barcelona MPL is analysed, applying the two-way transmittance method. First, a review of the literature on the two-way transmittance method which provides cirrus cloud retrievals like the cloud optical depth, the columnar cloud lidar ratio or the vertical profile of the particle backscatter coefficient was presented. The different approaches that have been developed along the year and the main advantages and disadvantages of this method were also explained. For example, one of the major advantages of this new approach of the method is that it is only necessary to assume a

Rayleigh zone both above and below the cirrus cloud, without making any priori optical and/or microphysical hypotheses about the cirrus cloud. Second, a simple mathematical development of the two-way transmittance method for ground-based and spaceborne lidar systems was proposed and was first illustrated for a cirrus cloud of the day 11-02-2019 at 02:03:50 UTC, in Barcelona, using measurements from the MPL and CALIOP lidars. The results of the two-way transmittance method fitted really well, obtaining a difference of COD for the same cirrus cloud or two different cirrus clouds in close proximity of 0.0054. Third, a set of criteria for cirrus clouds identification was established,

which consists of $T_{base}$ < -37ºC and CBH > 7 km, and was compared with the literature. After having carried out the identification of 367 high-altitude cirrus clouds, measured with the MPL in Barcelona, from November 2018 to September 2022, the two-way transmittance method has been applied correctly to the 55% of the all cases, because of the very noisy lidar signal or the non-guaranteed Rayleigh zone below and above the cirrus cloud. The cirrus geometrical and physical properties were: CT 1.8± 1.1 km, $T_m$ -51± 8 ºC, COD 0.36± 0.45, LR 30± 19 sr and LCDR 0.32± 0.13, with the highest occurrence in spring. It could be said that in the warmer seasons, opaque cirrus

are more frequent than visible cirrus. In addition, these properties were compared to the literature, obtaining similar properties in nearby latitudes, with a majority of visible and opaque cirrus clouds being present. It was also found that the occurrence of each cirrus depends on the weather pattern of each site and the lidar ratio seemed to have a generally increasing trend towards the poles and a decreasing trend towards the equator. Forth, it was found that the efficiency of the two-way transmittance method and the properties of the cirrus clouds were not dependent on the hour of day and their properties were analysed according to the COD. Resulting in that the subvisible cirrus clouds

were the highest, coldest and thinnest clouds; the visible cirrus clouds were the predominant and the opaque cirrus clouds were the lowest, warmest and thickest clouds in the whole cirrus dataset. It has also been seen that the cloud top height did not vary considerably depending on the type of cloud, since the cirrus clouds might reach to/near the tropopause, being its average height of 1.1±1 km during the cirrus scenes. The correlations between the different cirrus properties were then analysed and quantified for the first time, being the highest correlation $R^2$=0.26 between $T_{base}$ and COD. The analysis showed that the COD correlates positively with the cloud base temperature, lidar ratio and

linear cloud depolarization ratio and negatively with the cloud base height. This means that as the cloud base temperature decreases, the cloud base height increases, being characteristic of the troposphere and the cloud optical depth decreases. This fact is because clouds located at higher altitudes are formed from air masses with a lower water vapour content and, therefore, their geometric and optical thickness are smaller. On the other hand, the lidar ratio increases with increasing cloud optical depth, as the complexity and diversity of ice crystal shapes increases, due to collisions and turbulence. Lastly, the linear cloud depolarization ratio has a slightly positive tendency with the cloud optical

depth, because as the cloud optical depth increases, the number of ice crystals increases and, as a consequence, the randomly aggregation of ice crystals within the cloud occurs more frequently, becoming ice crystals rougher and consequently, depolarization increases. Finally, the dependence of LCDR on COD and LR was studied and it was concluded that cirrus clouds with LCDR values lower than 0.3 did not have liquid water, validating the cloud identification criteria proposed in this study. All these information presented in this work could be of great use for gaining a better understanding of the properties of cirrus clouds, their spatial distribution at the global scale and the key processes which govern cirrus formation and evolution. This study could also help development of new parameterizations of cirrus clouds to obtain



their optical, microphysical and radiative properties and development of cirrus clouds products obtained with spaceborne or ground-based lidar instruments.

*Data availability.* The MPLNET products are publicly available on the MPLNET website (https://mplnet.gsfc.nasa.gov/download_tool/) (MPLNET, 2023) in accordance with the data policy statement. The CALIPSO product is provided by the NASA Langley Research Center's
(LaRC) ASDC DAAC and is managed by the NASA Earth Science Data and Information System (ESDIS) project. NASA data are freely accessible and available on the Atmospheric Science Data Center website (https://asdc.larc.nasa.gov/) (NASA, 2023). Radiosoundings data are available upon request from the authors or Meteocat.

*Author contributions.* CGD prepared the automatic algorithm for the identification of cirrus clouds and the application of the two-way transmittance method for MPL and radiosonde data. CGD prepared the figures of the paper. MS, AC, CMP, ARG SL, JRL and EJW reviewed
different parts of the results. DCFSO took care of the maintenance of the MPL. CGD and MS prepared the paper, with contributions from all co-authors.

*Competing interests.* At least one of the (co-)authors is a member of the editorial board of Atmospheric Measurement Techniques.

*Acknowledgements.* The authors acknowledge the funding of this research by the Spanish State Agency for Research (AEI) for the project PID2019-103886RB-I00 and its support to ACTRIS ERIC. The authors also acknowledge the support of the European Commission through
the Horizon 2020 Research and Innovation Framework Programme projects ACTRIS IMP (grant agreement No 871115), ATMO-ACCESS (grant agreement No 101008004), GRASP-ACE (grant agreement No 778349) and REALISTIC (grant agreement No 101086690).





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
