# Peer review of "Geometrical and optical properties of cirrus clouds in Barcelona, Spain: Analysis with the two-way transmittance method of 4 years of lidar measurements"

_Atmospheric Measurement Techniques, 2023_

## Referee Comment (RC1)

This study makes use of five years of ground-based lidar measurements of cirrus geometrical and optical properties that are retrieved using the established two-way transmittance method over Barcelona. Optical and geometric properties are compared against other studies in a meta-analysis. Relationships among the different optical properties, and thermodynamic properties from radiosoundings, are examined.

The methodology is fairly sound but the authors should also consider in more detail the effects of conditional-sampling in their methodology as their method (two-way transmittance technique) is also only applicable to a non-random subset of the data.
The authors make conclusions about several relationships (e.g. latitudinal dependence of lidar ratio, and a relationship between depolarization ratio and cloud optical depth) that do not appear to be supported by the presented data. These conclusions should be reexamined. The authors also need to clearly distinguish between what is concluded based on this study, and their hypotheses about the causes of any observed relationships.
With these issues, I am left uncertain as to the novel contribution of this article to our understanding of cirrus properties or to our understanding of lidar remote sensing of cirrus. No new insight into the two-way transmittance technique is provided. Nor are the differences between the two-way transmittance method and the results of the operational MPLNET or CALIOP algorithms evaluated. The general conclusion seems to be that cirrus properties over Barcelona are similar to other studies. This is not without merit, but AMT does not seem to be the most appropriate venue for this manuscript in its present form, as AMT focuses on the development, intercomparison and validation of measurement techniques. As such, I recommend rejection of the manuscript.

I have made some suggestions to improve the manuscript and raise specific technical issues below.

Specific Comments:

Line 27:
The introduction is a little misleading beginning with Aerosol-radiation as well as aerosol-cloud interactions, despite neither being the subject of this paper. I suggest that the first paragraph be adjusted to begin by discussing the importance of cirrus as is done halfway through the paragraph.

Line 41:
I suggest the second paragraph omit the first two sentences and begin with "Cirrus clouds can form by different …"

Line 53:
I suggest that the third paragraph could be made stronger by being arranged in an argument that motivates this work as follows:
   1.  Ice cloud microphysics and their relationship to optical/radiative properties is complex.

2. Remote sensing of cirrus properties requires the assumption of a crystal habit or adoption of a particular empirical model, which complicates the results.
3. Lidar provide the ability to infer cloud optical depth etc. without making such assumptions.

This provides the same background but also more clearly motivates the importance and use of lidar remote sensing.

Line 68:
Lidar systems do not measure vertical profiles of extinction, in general, but in some cases can retrieve it.

Line 113:
Multiple scattering contributions do not depend only on the receiver field of view. The other relevant factors should be mentioned and additional references should be provided to justify this choice (e.g. Shcherbakov et al. 2022).

Line 234:
This is not the definition of the cloud optical depth. The optical depth is the vertical integral of the volume extinction coefficient. Definitions need to stay consistent to preserve meaning. This equation should be modified to explicitly include the multiple scattering correction, with the note that it is assumed to be negligible.

Equations:
Please use the same notation for integrals over range/altitude. The vertical coordinate variously appears as x, u, and z which is confusing.

Line 278:
This does not seem like a very precise convergence criterion.

Line 313:
I am not sure about these criteria. Does this eliminate the possibility of multiple layers of cirrus? Shouldn't we want to know the properties of both layers?

Line 322:
A success rate of 55% indicates that a significant fraction of data are omitted from the analysis. Any systematic reason for the omission of the data might substantially alter the resulting analysis. For example, it is stated at Line 318 that cases with high lidar ratio, typically with high levels of noise, are discarded. If this noise is caused by low signal strength due to strong attenuation (rather than noise in the lidar signal itself or solar noise), then this indicates a systematic sampling bias that should be discussed.
It is not clear whether the cirrus category in Figure 3 only includes the 203 cases, as a COD is derived, or whether the success of the two-way transmittance method is judged based on the lidar ratio. I suggest separating results into "non-cirrus, successful cirrus, failed cirrus" cases.
It was stated earlier that the two-way-transmittance test will fail for very optically thin clouds (i.e. subvisible). Some justification is required for why the statistics of subvisible cirrus should

be treated as representative. Uncertainties should be propagated to establish the precision of these retrievals.

Figure 4:
Again, the daytime/nighttime contrast should be partitioned by retrieval failure or success.

Figure 5a: The bins are not particularly clear. I suggest logarithmically spaced bins as well.

Table 3:
The meaning of the quantity after the +/- needs to be defined. Is this the standard deviation? Or the standard error in the mean?

Line 430:
I would disagree with the conclusion that the lidar ratio has a generally increasing trend towards the poles. Instead, my conclusion would be "the variability at different sites appears negligible relative to the variability at each site."

Line 452:
It needs to be clarified whether this correlation is between COD and the other cirrus properties or between $\log_{10}(COD)$.

Figure 6:
The grey shading does not appear to be the 95% confidence intervals of the linear regression. I would expect uncertainty in the slope of the regression to produce diverging bounds on the relationship in a "><" shape, unless the standard error in the slope is negligible compared to the standard error in the intercept, which I would not expect to be the case for the shown data.

Line 470:
My reading of Figure 6 (bottom) is opposite to the authors in that there is no significant relationship between LCDR and COD. The r-squared value is 0.03.
The reference entry for Chen et al. 2002 appears to be incorrect.
What I presume to be the correct reference (below) suggests a decreasing relationship between LCDR and COD. This study is distinct in that there is no significant relationship.

Figure 7:
The caption refers to a known range of lidar ratio for cirrus clouds being less than 40 sr. Some references are required for this. The authors should bear in mind that in situ measurements of lidar ratio are not column averaged, while what is reported here is an effective column lidar ratio.

The authors should comment on the possibility that the MPLNET cloud classification used to define cloud in this study is misclassifying aerosol as cloud and that that contributes to the low depolarization ratios.

Line 480:
Should the thermodynamics not also be a major indicator here? How many cirrus clouds even have cloud-base temperatures that are above the homogeneous nucleation temperature?

Line 482:
I suggest focusing on the warmest temperature within the cloud (i.e. cloud base) for this determination, rather than mentioning altitude.

Line 499:
"It could be said" is vague. Just state the result.

Line 502:
No weather patterns were examined, so this should not be a conclusion. Rather, it is a hypothesis about the differences between sites. The latitudinal dependence does not seem significant.

Line 507:
The average height of cirrus is probably not 1.1 km.

Lines 510-517:
This is too long for the conclusions and repeats information. Moreover, several hypotheses about the cause of the results are presented as strong conclusions.
For example, the lidar ratio increases with COD because of turbulence). Turbulence was not measured and this attribution cannot be concluded.
The linear depolarization does not appear to have any relationship. Certainly, there isn't any cause to attribute any relationship to increases in aggregation, as opposed to e.g. micro-facet roughness of the crystals.

References:
Wei-Nai Chen, Chih-Wei Chiang, and Jan-Bai Nee, "Lidar ratio and depolarization ratio for cirrus clouds," Appl. Opt. **41**, 6470-6476 (2002)

Shcherbakov, V., Szczap, F., Alkasem, A., Mioche, G., and Cornet, C.: Empirical model of multiple-scattering effect on single-wavelength lidar data of aerosols and clouds, Atmos. Meas. Tech., 15, 1729–1754, https://doi.org/10.5194/amt-15-1729-2022, 2022

---

## Author Comment (AC1)

Dear reviewer,

I attach in this document the answers to your comments. But first of all, I would like to thank you for spending time with the review of this manuscript. The answers are in blue and the references made to the lines are made with respect to the new version of the manuscript.

**Line 27:**

The introduction is a little misleading beginning with Aerosol-radiation as well as aerosol-cloud interactions, despite neither being the subject of this paper. I suggest that the first paragraph be adjusted to begin by discussing the importance of cirrus as is done halfway through the paragraph.

I accept your suggestion and now the paragraph starts directly with the topic of cirrus clouds. I copy the beginning of the paragraph.

"The radiative effect of high-altitude cirrus clouds plays a fundamental role in the global radiation budget (Liou, 1986; Lolli et al., 2017). Despite that, they have been designated as poorly understood by (IPCC, 2021) because of a lack of knowledge of their dynamic, microphysical and radiative properties. Indeed, cirrus cloud critical role in the climate comes from the fact … "

**Line 41:**

I suggest the second paragraph omit the first two sentences and begin with "Cirrus clouds can form by different …"

The definition of Met Office was showed because we wanted to emphasize the composition of cirrus clouds. However, I accept your suggestion and now the Met Office definition has been removed. I copy the beginning of the paragraph.

"Cirrus clouds are mainly composed ice crystals and can form through different atmospheric mechanisms, giving rise to cirrus clouds with different physical, geometrical and optical properties."

**Line 53:**

I suggest that the third paragraph could be made stronger by being arranged in an argument that motivates this work as follows:

1. Ice cloud microphysics and their relationship to optical/radiative properties is complex. 2. Remote sensing of cirrus properties requires the assumption of a crystal habit or adoption of a particular empirical model, which complicates the results.

3. Lidar provide the ability to infer cloud optical depth etc. without making such assumptions.

This provides the same background but also more clearly motivates the importance and use of lidar remote sensing.

I accept your suggestion and after the explanation of the different ways of calculating the radiative properties of cirrus clouds, the focus has been changed to provide more motivation. I copy the paragraph.

"Ice cloud microphysics and their relationship to optical/radiative properties is complex. Cirrus clouds can be characterized by some key parameters such as the mid-cloud altitude and temperature, cloud extinction coefficient, cloud optical depth, lidar ratio (LR) or linear cloud depolarization ratio (LCDR). While the LR and LCDR are related with the microphysical properties of the ice crystals contained in cirrus clouds,

such as their shape and/or orientation, the mid-cloud altitude and temperature as well as the cloud extinction coefficient play an important role in determining the cloud radiative properties. Up to the present date, there is no exact theoretical solution for scattering and absorption by non-spherical ice particles (Liou and Takano, 1994). Nevertheless, scattering models for cirrus clouds have been developed, such as (Baran et al, 2009, 2011a, b) which relates the cirrus ice water content and mid-cloud temperature with its extinction coefficient and radiative properties. Alternatively, (Heymsfield et al., 2014; Dolinar et al., 2022) propose a relationship between the ice water content with the extinction coefficient and the cloud temperature with the effective geometric diameter of ice crystals. From these properties, the cirrus cloud radiative properties can be calculated with the (Fu et al., 1998, 1999) parametrizations. These and other ways of obtaining the radiative properties of cirrus clouds have several points in common, such as the need to calculate the cloud extinction, where the application of remote sensing is essential, or the assumption of the ice crystal shape distribution in empirical models, further complicating the results."

**Line 68:**

Lidar systems do not measure vertical profiles of extinction, in general, but in some cases can retrieve it.

Right, lidar systems do not measure directly vertical profiles of extinction, but they can retrieve them. Therefore, the verb measure has been changed to retrieve.

**Line 113:**

Multiple scattering contributions do not depend only on the receiver field of view. The other relevant factors should be mentioned and additional references should be provided to justify this choice (e.g. Shcherbakov et al. 2022).

Right, a more rigorous explanation has been made in line 206. I copy the explanation.

"The multiple scattering factor, η, is introduced by (Platt, 1973, 1979). The multiple scattering effect depends on laser beam divergence, receiver field of view, the distance between the light source and the scattering volume (Wandinger, 1998; Wandinger et al., 2010; Shcherbakov et al., 2022). In this study the multiple scattering effect is considered negligible for lidar signal measured by the MPL system due to its narrow field of view, the mean distance between cirrus clouds and the MPL, the small cirrus cloud optical depth (generally COD < 0.3) and the magnitude of cirrus cloud extinction ($\alpha$p < 1 km−1) retrieved (Campbell et al., 2002; Lewis et al., 2016; Shcherbakov et al., 2022)."

**Line 234:**

This is not the definition of the cloud optical depth. The optical depth is the vertical integral of the volume extinction coefficient. Definitions need to stay consistent to preserve meaning. This equation should be modified to explicitly include the multiple scattering correction, with the note that it is assumed to be negligible.

Right, the notation of the volume extinction coefficient has been changed and in line 203, the volume particle extinction coefficient has been denoted as the volume effective extinction coefficient corrected by multiple scattering errors, whose mathematical expression is $\alpha_p = \eta\,\alpha_{ef}$, being $\alpha_{ef}$ the volume effective extinction coefficient, which is measured by the MPL system.

**Equations:**

Please use the same notation for integrals over range/altitude. The vertical coordinate variously appears as x, u, and z which is confusing.

Ok, the notation for integrals was unified with the variable z.

**Line 278:**

This does not seem like a very precise convergence criterion.

The convergence criterion is 1 sr and is very precise. No convergence problems have been found in this respect. In fact, this iterative algorithm with the convergence criterion of 1 sr has an average of 3 iterations.

**Line 313:**

I am not sure about these criteria. Does this eliminate the possibility of multiple layers of cirrus? Shouldn't we want to know the properties of both layers?

Yes, we have removed the possibility of multi-layers of cirrus clouds, because with these conditions it is not possible to have a molecular region above and below each cirrus, in order to be able to apply the two-way transmittance method.

In the case of a multi-layer cirrus cloud, if the distance between clouds is less than 1 km, it is analyzed as if they were one cloud in total, not as several cirrus clouds in close proximity. On the contrary, it could not be considered that below the upper cirrus in the vertical profile there is a Rayleigh zone. The distance required for normalization in Rayleigh zones is 5 km above the cirrus cloud and 1 km below.

I would like to know the properties of all the cirrus layers but with the two-way transmittance method developed in this manuscript it is not possible.

**Line 322:**

A success rate of 55% indicates that a significant fraction of data are omitted from the analysis. Any systematic reason for the omission of the data might substantially alter the resulting analysis. For example, it is stated at Line 318 that cases with high lidar ratio, typically with high levels of noise, are discarded. If this noise is caused by low signal strength due to strong attenuation (rather than noise in the lidar signal itself or solar noise), then this indicates a systematic sampling bias that should be discussed. It is not clear whether the cirrus category in Figure 3 only includes the 203 cases, as a COD is derived, or whether the success of the two-way transmittance method is judged based on the lidar ratio. I suggest separating results into "non-cirrus, successful cirrus, failed cirrus" cases. It was stated earlier that the two-way-transmittance test will fail for very optically thin clouds (i.e. subvisible). Some justification is required for why the statistics of subvisible cirrus should be treated as representative. Uncertainties should be propagated to establish the precision of these retrievals.

I will summarize the number of cirrus cases found. In this manuscript, only 1025 days have been analyzed, at 00 and 12 UTC, so there are 2050 cases. Of these 2050 cases, a cloud has been detected with MPLNET products in 1019 cases (49.7%). Of these 1019 cloud cases, at least one cirrus cloud has been detected in 367 cases (36% of sub-dataset of 1019 cloud cases). On the other hand, of these 367 cases, 164 cases could not be correctly applied to the two-way transmittance method. In line 321, the detection of errors in the application of the two-way transmittance method is explained in more detail. I copy the paragraph.

"Of these 367 cases, the two-way transmittance method has only been correctly applied to 203 cases, denoted as "successful" 320 cirrus. Of the 164 cases of cirrus clouds to which the two-way transmittance method could not be correctly applied, denoted as "failed" cirrus, in 29%, the Rayleigh zone above and below the cirrus cloud could not be guaranteed, in 46% a negative COD was calculated and in 25% a LR higher than 100 sr was estimated. Of the "failed" cirrus cases for which the Rayleigh zone above and below

the cirrus cloud could be guaranteed, in 92% of the cases, the cirrus had a very small lidar signal peak and in 8% of the cases, although the lidar signal peak associated to the cirrus cloud was noticeable, the signal was excessively noisy."

A deeper analysis of "failed" cirrus clouds has been also developed. In Figure 3a, the temporal distribution of "failed" cirrus clouds was added and Figure 3b has been better explained in its figure caption.

**Figure 4:**

Again, the daytime/nighttime contrast should be partitioned by retrieval failure or success.

It was added in line 365.

"The efficiency of the two-way transmittance method does not seem to be affected considerably, since the success rates of this method for cirrus clouds during daytime (62%) and nighttime (51%) are similar."

**Figure 5a:**

The bins are not particularly clear. I suggest logarithmically spaced bins as well.

Right, the figure was changed and for sub-visible and visible cirrus, a logarithmic grid was used. I attach the new figure.

[Figure]

**Figure 5.** Probability distribution of (left) cloud optical depth, (center) effective column lidar ratio and (right) linear cloud depolarization ratio, calculated using the two-way transmittance method, from 2018 to 2022 in Barcelona. The rectangles in the upper right-hand corners show average values and standard deviations of the distributions. The left figure has a logarithmic grid to show the sub-visible and visible cloud groups.

**Table 3:**

The meaning of the quantity after the +/- needs to be defined. Is this the standard deviation? Or the standard error in the mean?

Yes, they are mean values and standard deviations. It was changed in line 431.

"**Table 3.** Average and standard deviation values of characteristics of cirrus clouds of ground-based lidar observations, reported in literature. The optical properties have been calculated at 532 nm. Where N is the number of cirrus clouds identified and (%) its percentage with respect to the total number of clouds. The occurrence of SVC, VC and opaque cirrus clouds are made on the number of cirrus N. (a) Tm values have been manually calculated from values of temperature at cloud and top heights, shown in the paper. (b) The geometrical properties show are from an annual average and the optical properties are obtained by the two-way transmittance method applying a multiple scattering correction. (c) The optical properties are calculated at 355 nm."

**Line 430:**

I would disagree with the conclusion that the lidar ratio has a generally increasing trend towards the poles. Instead, my conclusion would be "the variability at different sites appears negligible relative to the variability at each site."

I see a positive trend of the lidar ratio towards the poles, but I agree that there is a large variability at each site. So I have changed the conclusion to the following: "the effective column lidar ratio seems to have a generally increasing trend towards the poles, but no conclusion can be drawn, since the variability at different sites appears negligible relative to the variability at each site."

**Line 452:**

It needs to be clarified whether this correlation is between COD and the other cirrus properties or between log10(COD).

Right, it was changed in line 466.

**Figure 6:**

The grey shading does not appear to be the 95% confidence intervals of the linear regression. I would expect uncertainty in the slope of the regression to produce diverging bounds on the relationship in a "x" shape, unless the standard error in the slope is negligible compared to the standard error in the intercept, which I would not expect to be the case for the shown data.

Right, only the interception error was considered. It has already been changed. I attach the new figure.

[Figure]

**Figure 6.** Logarithmic dependence of the cloud optical depth with (a) cloud base temperature, (b) cloud base height, (c) effective column lidar ratio and (d) linear cloud depolarization ratio, for cirrus cases from 2018 to 2022 in Barcelona. The solid black line is the linear regression that has been calculated between the variables and the grey shading with the dash dotted black lines are the 95% confidence limit of the linear regression. The $R^2$ coefficients are (a) 0.26, (b) 0.19, (c) 0.17 and (d) 0.03.

**Line 470:**

My reading of Figure 6 (bottom) is opposite to the authors in that there is no significant relationship between LCDR and COD. The r-squared value is 0.03. The reference entry for Chen et al. 2002 appears to be incorrect. What I presume to be the correct reference (below) suggests a decreasing relationship between LCDR and COD. This study is distinct in that there is no significant relationship.

Right, I changed the reference of Chen et al., 2002. Sorry for the mistake. The correct reference is "Chen, W.-N., Chiang, C.-W., and Nee, J.-B.: Lidar ratio and depolarization ratio for cirrus clouds, Appl. Opt. 41, 6470-6476, 2002.".

It is true that despite the clear visualization of the positive correlation between LCDR and COD, their r-squared is very low. Therefore, that statement has been changed in the text. (Line 485)

"Likewise, the linear cloud depolarization ratio has a slightly positive tendency with the cloud optical depth, which is negligible because of its low R-squared of 0.03. Moreover, (Chen 485 et al., 2002) found an opposed tendency. Despite that, a positive tendency between LCDR and COD could make sense due to the fact that as the COD increases, the number of ice crystals increases and, as a consequence, the randomly aggregation of ice crystals within the cloud occurs more frequently. As the ice crystals increase in size, they become rougher and consequently, depolarization increases (Yang et al., 2000)."

**Figure 7:**

The caption refers to a known range of lidar ratio for cirrus clouds being less than 40 sr. Some references are required for this. The authors should bear in mind that in situ measurements of lidar ratio are not column averaged, while what is reported here is an effective column lidar ratio.

The authors should comment on the possibility that the MPLNET cloud classification used to define cloud in this study is misclassifying aerosol as cloud and that that contributes to the low depolarization ratios.

In Figure 7, the lidar ratio interval that was considered as normal for cirrus clouds was 10-40 sr, giving a margin of 10 sr to the range of 20-30 sr which agrees with (Sassen and Comstock, 2001; Yorks et al., 2011; Josset et al., 2012; Garnier et al., 2015; Cordoba-Jabonero et al., 2017). On the other hand, for linear cloud depolarization ratio values, the established range was 0.3-0.5, according to (Sassen, 2005; Giannakaki et al., 2007; Kim et al., 2018; Hu et al., 2021).

I accept the suggestion of the lidar ratio notation and it was changed by effective column lidar ratio. I comment our confidence in MPLNET's products and its procedures on the line 340. The misclassification aerosol as cloud is possible, but I am confident in the reliability of the products, in the absence of evidence to the contrary.

**Line 480:**

Should the thermodynamics not also be a major indicator here? How many cirrus clouds even have cloud-base temperatures that are above the homogeneous nucleation temperature?

No, because the goal is to analyze the existence of liquid water content on cirrus clouds depending their optical properties. Specifically, we focus on cirrus clouds that have optical properties that do not fit well with the literature. For this reason, the boxes are fixed in LR < 10 sr and LR > 40 sr and LCDR < 0.3. (See section 4.2 Cirrus optical properties).

It is a good question and I added the answer to the discussion. I copy the paragraph.

"On the other hand, visible and opaque cirrus clouds control the blue sector. The percentage of cirrus found in this area is 12%. As one cloud type does not predominate, the geometrical properties of this subgroup have been analysed, showing an average of cloud base temperature of -41.32±8.62 ºC, being lower to the homogeneous nucleation temperature of -38.15 ºC, (Tanaka and Kimura, 2019) and making the presence of aqueous content in these cirrus clouds impossible. However, eight cases have been found with a temperature above -38.15ºC and an average cloud base height of 7.91±0.68 km. Therefore, in these eight cases the presence of liquid water cannot be ruled out. Except for these 8 cases, the validation of the cloud identification criteria proposed in this study can be successfully concluded."

In this case, thanks to your question, I have rethought my analysis. Now the analysis has been changed and is based on the cloud base temperature. It is true that if the cloud base temperature is higher than the homogeneous nucleation temperature, the presence of liquid water in the cloud cannot be ruled out. Therefore, it has been concluded that, except for 8 cases, the presence of aqueous content in the rest of the cirrus clouds analysed has been ruled out, thus validating the identification of cirrus clouds proposed in this study.

**Line 482:**

I suggest focusing on the warmest temperature within the cloud (i.e. cloud base) for this determination, rather than mentioning altitude.

I accept your suggestion and now the analysis has been changed and is based on the cloud base temperature. I explained in the previous comment.

**Line 499:**

"It could be said" is vague. Just state the result.

Right, it was changed.

**Line 502:**

No weather patterns were examined, so this should not be a conclusion. Rather, it is a hypothesis about the differences between sites. The latitudinal dependence does not seem significant.

Right, the variability at each site is high.

**Line 507:**

The average height of cirrus is probably not 1.1 km.

Right, it is 11 km. Sorry for the mistake.

**Lines 510-517:**

This is too long for the conclusions and repeats information. Moreover, several hypotheses about the cause of the results are presented as strong conclusions. For example, the lidar ratio increases with COD because of turbulence). Turbulence was not measured and this attribution cannot be concluded. The linear depolarization does not appear to have any relationship. Certainly, there isn't any cause to attribute any relationship to increases in aggregation, as opposed to e.g. micro-facet roughness of the crystals.

Ok, I accept the suggestions and I changed the conclusion. I copy the entire conclusions.

[revised manuscript text omitted]

---

## Author Comment (AC2)

Dear Dr. Schumann,

I would first like to thank you for your interest in this manuscript. We cannot in fact guarantee the exclusion of contrails nor the number of contrails in the sample analyzed. Thank you very much for the information provided on cirrus contrails, I think it is worth to add information about cirrus contrails to the introduction of the manuscript. Finally, no radiosonde humidity data have been used as they were not necessary for the analysis carried out.

I copy the part of the introduction where I mention the cirrus contrails.

"Cirrus clouds can also be triggered from aircraft contrails. These contrails are caused by aircraft engine exhaust, primarily water, which turns into ice crystals at low temperature. Cirrus contrails are often formed in persistent humidity conditions (Schumann, 1996; Schumann and Heymsfield, 2007; Schumann et al., 2021; Li et al., 2023). Their lifetimes sometimes reach several hours and their spatial extension may evolve up to 10 km in width and between 0.5 and 1.5 km in depth. Moreover, cirrus contrails from several aircraft may often overlap and form together a larger contrail cirrus cloud, making it more difficult to distinguish from other cirrus."

Thank you for your contribution,

Best wishes,

Cristina Gil.

---

## Author Comment (AC4)

Dear reviewer,
I attach in this document the answers to your comments. But first of all, I would like to thank you for spending time with the review of this manuscript. The answers are in blue and the references made to the lines are made with respect to the new version of the manuscript.

**General comment**
According to authors, the detection of the cirrus clouds is made to fulfill two criteria, one about the temperature at the cloud top height and the other about the cloud base height. What about the signal to noise ratio, before applying the cirrus detection? The SNR should also be checked.

The detection of cirrus clouds as mentioned is performed with the cloud top temperature ($T_{top}$) and the cloud base height (CBH). For this, first the cloud cases are identified as 1-min vertical profiles where the MPLNET CLD product has a valid cloud base and top value. Then, using the radiosonde data, the cloud top temperature is estimated. Once the cloud top temperature and the cloud base height are obtained, it is checked if they fulfill certain conditions (CBH > 7 km; $T_{top}$ < - 37ºC). If so, these clouds are identified as cirrus clouds.

For the cirrus cloud identification, we do not check the SNR, because we use the cloud base and top height variables of the MPLNET CLD product. MPLNET calculates these variables considering the SNR of the lidar signal (Lewis et al., 2016). Although the two-way transmittance method does not work successfully with noisy lidar signals.

In lines 352-354 the authors explain the conditions for the cloud identification with MPLNET CLD product and express their confidence in MPLNET's products and procedures.

What about the smoothing that the authors apply to the lidar signal?

The NRB signal profiles are temporally averaged to represent the merged cloud scenes. This average of the NRB signal is done according to the averaging that MPLNET has done to the 1-min profiles, being indicated in the variable "time_average" of the MPLNET CLD product. This multi-temporal averaging scheme is used to improve high-altitude cloud detection under conditions of a weak signal-to-noise ratio (Lewis et al., 2016, 2020). It is explained in line 113.

How can authors explain the detection of cirrus clouds with depolarization values less than 0.1? Figure 5 (right) is depicting linear cloud depolarization ratio in the bin between 0 and 0.1? Have you checked the SNR of these causes? The depolarization values are really surprising for cirrus clouds. Moreover, the 1-min temporal resolution could have restricted the accuracy of the depolarization ratio.

Yes, the explanation has been improved and can be found on line 439. I copy the paragraph. "The linear cloud depolarization ratio is typically between 0.3-0.5 (54%), with an average of 0.32, which is in agreement with (Sassen, 2005; Giannakaki et al., 2007; Kim et al., 2018; Hu et al., 2021). The lowest values of the linear cloud depolarization ratio may be due to a tendency of horizontal orientation of the ice crystals or a very thin or multi-layered cloud (Hu et al., 2009). It is mentioned above that in this study if there is another cloud lower, less than 1 km away, the two clouds are merged and treated as one cloud layer."

As explained in the previous question, in this study the multi-temporal averaging scheme is used to improve high-altitude cloud detection under conditions of a weak signal-to-noise ratio (Lewis et al., 2016,

2020). In this case, there are 13 merged cirrus scenes with depolarization values lower than 0.1. Of these 13 merged cirrus scenes, 6 are cloud scenes averaged over 1 min, 4 cases are averaged over 5 min and 3 are averaged over 21 min.

Do the authors apply any integration for the cloud retrievals?

No, we make averages of a half-cloud vertical profiles, centred at the maximum peak to calculate cirrus cloud retrievals. (Line 302)

Authors claim that "For example, one of the major advantages of this new approach of the method is that it is only necessary to assume a Rayleigh zone both above and below the cirrus cloud, without making any priori optical and/or microphysical hypotheses about the cirrus Cloud". The authors should provide more details and even calculations about the errors introducing in their statistics with this approach in detecting cirrus clouds.

No errors were calculated, but on request a section with the statistical study of the cirrus cloud retrieval errors have been added. The methodology is in Section 3.5 and the statistics of the error of the retrievals applied to the database is in Section 5.2.

"After the calculation of the cirrus clouds optical retrievals, their associated errors have been estimated. Where the COD, LR and LCDR errors have been calculated for each cirrus cloud scene with the classical error propagation equations (Ku, 1966). Similarly to the calculation of the LR and LCDR, their errors have been estimated by performing the average on half-cloud, centred at the maximum peak. In addition, the LR error has been calculated as the maximum possible error, since only the first iteration has been considered in its calculation. As the classical error propagation equations have been used, it has been necessary to establish the errors of some variables such as the temperature and pressure of the radiosondes, being $\triangle T = 0.2ºC$ and $\triangle P = 0.5hPa$ (Servei Meteorològic de Catalunya, 2005). The MDR error has been quantified as 3.5% of its value (Behrendt and Nakamura, 2002). The NRB and VDR errors have been assumed to be the NRB and VDR uncertainties from MPLNET NRB product."

"After having shown the probability distributions and the mean and standard deviation values of the cirrus clouds optical retrievals, the basic statistical values of their associated errors are presented in Table 3.

| Variables | Min | Mean | Median | Std | Max |
|:---:|:---:|:---:|:---:|:---:|:---:|
| COD | 0.04 | 0.16 | 0.11 | 0.20 | 1.54 |
| LR (sr) | 0.00* | 0.28 | 0.06 | 0.84 | 7.83 |
| LCDR | 0.01 | 0.18 | 0.08 | 0.31 | 2.06 |

**Table 3.** Minimum, mean, median, standard deviation and maximum values of the COD, LR and LCDR errors for cirrus cases from 2018 to 2022 in Barcelona. *Zero values are not exactly null, but if rounded to the second hundredth they can be considered null.

Table 3 shows that the error of the COD is 0.16±0.20 with a maximum value of 1.54, being considerably high for sub-visible cirrus clouds (COD < 0.03), but reasonable for visible and opaque cirrus clouds. In addition, the maximum COD error found is lower than the maximum COD calculated. The LR error is

0.28±0.84 sr with a maximum value of 7.83 sr. If it is compared to its magnitude (30±19 sr; see Figure 5) is negligible in most cases. On the contrary, the LCDR error is 0.18±0.31, which is considerable for the lowest values, since the LCDR ranges between 0 and 1. In addition, a maximum LCDR error of 2.06 has been calculated, being greater than unity. This error is so large due to the uncertainty associated with this vertical profile of volume depolarization ratio."

Line 17. The authors claim that: «Together with results from other sites, a possible latitudinal dependence of lidar ratio is detected: the lidar ratio increases with increasing latitude. » This sentence is not supported from the study retrievals. It must be removed.

Right, it was changed and it is explained in line 466. We see a positive trend of the lidar ratio towards the poles, but we admit that there is a large variability at each site. So, we have changed that conclusion to the following: "the effective column lidar ratio seems to have a generally increasing trend towards the poles, but no conclusion can be drawn, since the variability at different sites appears negligible relative to the variability at each site."

How is the calibration of the polar and cross-polar channel made?

MPLNET polarized MPLs use a ferroelectric liquid crystal to alternate polarization states between linear and elliptically emitted laser pulses, and the data are calibrated using the optical specifications of key optical components and determination of the offset angle between the crystal's primary fast axis and the lidar's fast axis (Welton et al., 2018). The lidar data are processed with the same procedure as used for the older neumatic liquid crystal design (Flynn et al., 2007) to calculate the volume depolarization ratio.

In the equation #6, the η factor is equal to 1 in your study. You should state this assumption.

Right, it was changed and not the multiple scattering effects are better explained in line 207.

"The multiple scattering factor, η, is introduced by (Platt, 1973, 1979). The multiple scattering effect depends on laser beam divergence, receiver field of view, the distance between the light source and the scattering volume (Wandinger, 1998; Wandinger et al., 2010; Shcherbakov et al., 2022). In this study the multiple scattering effect is considered negligible for lidar signal measured by the MPL system ($\eta = 1$) due to its narrow field of view, the mean distance between cirrus clouds and the MPL, the small cirrus cloud optical depth (generally COD < 0.3) and the magnitude of cirrus cloud extinction ($\alpha_p < 1\ km^{-1}$) retrieved (Campbell et al., 2002; Lewis et al., 2016; Shcherbakov et al., 2022)."

How CALIPSO accounts for the multiple scattering effect of the ice crystals?

At the first, the multiple scattering effect for CALIPSO signal was ignored but it was changed and now the multiple scattering effects are considered in CALIPSO lidar signal. In line 254, the value of the multiple scattering factor is explained.

"The multiple scattering effect cannot be neglected for spaceborne lidar signals because of the distance between the satellite and the cirrus clouds. For this reason, η is assumed constant throughout the cloud layer with a value of 0.6, as in the version 3 of CALIOP algorithm (Garnier et al., 2015)."

Therefore, in the case study where the two-way transmittance method is applied to the CALIPSO lidar signal, the effect of multiple scattering is considered and its COD is changed to 0.2547.

[Figure]

**Figure 1.** Application of the two-way transmittance method for (left) MPLNET and (right) CALIPSO data, for the case 11/02/2019 at 02:03:50 UTC. The height zb(zt) is the altitude corresponding to 0.2 km above (below) cloud top (base) height.

Table 1 and its description is now included in the section of results. It should be moved before Section 4.

Ok, we accept your suggestion and we have changed that section to a separate section called "Criteria for cirrus cloud identification" and we placed it before section 4.

I propose a reconstruction of the text. The title of Section 4.4 "Cirrus correlation" is misleading. It must be changed.

Ok, so as not to make a mistake in interpretation of that section, we have accepted your suggestion and renamed it as "Discussion".

Line 80-85. The novelty of the work needs to be discussed and detailed.

The introduction section was slightly changed to emphasize the novelties that this study brings in the current context. The conclusions have also been modified to emphasize the contribution of this study compared to the literature. For example, the possible causes of error found in the application of the two-way transmittance method and the errors associated to cirrus cloud retrievals have been studied.

**Specific comments**

Page 2 line 44-45. Reference is missing.

The introduction section was greatly changed. In fact, the following paragraph that you mention has been deleted.

"The Met Office (the national meteorological service for the United Kingdom; https://www.metoffice.gov.uk/) defines cirrus clouds as "short, detached, hair-like clouds found at high altitudes"."

Page 27, Line 590. Replace "Depolarizationratio" with "Depolarization ratio".

Right, thanks. It was changed.

**References:**

Campbell, J. R., Hlavka, D. L., Welton, E. J., Flynn, C. J., Turner, D. D., Spinhirne, J. D., Stanley, V., Iii, S., and Hwang, I. H.: Full-Time, Eye-Safe Cloud and Aerosol Lidar Observation at Atmospheric Radiation Measurement Program Sites: Instruments and Data Processing, J. Atmos. Ocean. Technol., 19, 2002.

Flynn, C. J., Mendoza, A., Zheng, Y., and Mathur, S.: Novel polarization-sensitive micropulse lidar measurement technique, Opt. Express, 15, 2785–2790, 2007.

Garnier, A., Pelon, J., Vaughan M. A., Winker D. M., Trepte, C. R., and Dubuisson, P.: Lidar multiple scattering factors inferred from CALIPSO lidar and IIR retrievals of semi-transparent cirrus cloud optical depths over oceans, Atmos. Meas. Tech., 8, 2759–2774, 615 doi:10.5194/amt-8-2759-2015, 2015.

Ku, H. H.: Notes on the use of propagation of error formulas, J. Res. Natl. Bur. Stand. Sect. C Eng. Instrum., vol. 70C, no. 4, p. 263, 1966.

Lewis, J. R., Campbell, J. R., Stewart, S. A., Tan, I., Welton, E. J., and Lolli, S.: Determining cloud thermodynamic phase from the polarized Micro Pulse Lidar, Atmos. Meas. Tech., 13(12), 6901–6913, doi: 10.5194/amt-13-6901-2020, 2020.

Lewis, J. R., Campbell, J. R., Welton, E. J., Stewart, S. A., and Haftings, P. C: Overview of MPLNET version 3 cloud detection. *JTECH, 33*(10), 2113–2134. https://doi.org/10.1175/JTECH-D-15-0190.1, 2016.

Servei Meteorològic de Catalunya, Departament de Medi Ambient i Habitatge, Generalitat de Catalunya: El radiosondatge 3: una anàlisi de l'atmosfera, Valant 2003, S.L. (1ª ed.), https://static-m.meteo.cat/wordpressweb/wp-content/uploads/2014/11/18120559/Radiosondatge.pdf, 2005.

Shcherbakov, V., Szczap, F., Alkasem, A., Mioche, G., and Cornet, C.: Empirical model of multiple-scattering effect on single-wavelength lidar data of aerosols and clouds, Atmos. Meas. Tech., 15, 1729–1754, https://doi.org/10.5194/amt-15-1729-2022, 2022.

Wandinger, U., Tesche, M., Seifert, P., Ansmann, A., Müller, D., and Althausen, D.: Size matters: Influence of multiple scattering on CALIPSO light-extinction profiling in desert dust, Geophys. Res. Lett., 37(10), 1-5. https://doi.org/10.1029/2010GL042815, 2010.

Wandinger, U.: Multiple-scattering influence on extinction and backscatter coefficient measurements with Raman and high-spectral resolution lidars, Appl. Opt., 37(3), 417. https://doi.org/10.1364/ao.37.000417, 1998.

Welton, E.J., Stewart, S.A., Lewis, J.R., Belcher, L.R., Campbell, J.R., and Lolli, S: Status of the NASA Micro Pulse Lidar Network (MPLNET): Overview of the network and future plans, new Version 3 data products, and the polarized MPL. EPJ Web of Conferences, 176, https://doi.org/10.1051/epjconf/201817609003, 2018.